# Coupling of ribosome biogenesis and translation initiation in human mitochondria

Marleen Heinrichs[1,2,9], Anna Franziska Finke ⬥[3,4,9], Shintaro Aibara[5], Angelique Krempler[1], Angela Boshnakovska ⬥[3], Peter Rehling ⬥[2,3,6,7,8], Hauke S. Hillen ⬥[2,3,4,8] ✉ & Ricarda Richter-Dennerlein ⬥[1,2,8] ✉

Biogenesis of mitoribosomes requires dedicated chaperones, RNA-modifying enzymes, and GTPases, and defects in mitoribosome assembly lead to severe mitochondriopathies in humans. Here, we characterize late-step assembly states of the small mitoribosomal subunit (mtSSU) by combining genetic perturbation and mutagenesis analysis with biochemical and structural approaches. Isolation of native mtSSU biogenesis intermediates via a FLAG-tagged variant of the GTPase MTG3 reveals three distinct assembly states, which show how factors cooperate to mature the 12S rRNA. In addition, we observe four distinct primed initiation mtSSU states with an incompletely matured rRNA, suggesting that biogenesis and translation initiation are not mutually exclusive processes but can occur simultaneously. Together, these results provide insights into mtSSU biogenesis and suggest a functional coupling between ribosome biogenesis and translation initiation in human mitochondria.

Ribosomes are large macromolecular RNA-protein complexes that synthesize proteins in a highly efficient and accurate manner. Human cells contain ribosomes in the cytosol, but also within mitochondria, where they synthesize the essential core subunits of the oxidative phosphorylation (OXPHOS) system. Defective mitochondrial ribosomes (mitoribosomes) lead to OXPHOS deficiency and thus to a decline in cellular energy production, ultimately leading to severe mitochondrial disorders (mitochondriopathies)[1]. To form functional ribosomes, the assistance of auxiliary factors is required that mediate RNA processing, modification and folding, recruit and guide ribosomal proteins, and that facilitate molecular switches by releasing other factors. Mitoribosomes and their assembly factors are evolutionarily related to the prokaryotic translation system. However, their structures and assembly pathways differ substantially[2–5]. The 55S human mitoribosome has a higher protein and reduced RNA content compared to its bacterial counterpart. It is composed of a 39S large subunit (mtLSU) containing the 16S rRNA, tRNA^Val and 52 proteins (MRPs), and a 28S small subunit (mtSSU) comprising the 12S rRNA and 30 MRPs. While all MRPs are encoded in the nucleus, synthesized in the cytosol and imported into mitochondria, the rRNA is encoded by the mitochondrial genome (mtDNA), transcribed as part of a polycistronic transcript and processed by mitochondrial RNase P and Z[6–9]. Due to these differences in composition and its dual genetic nature, mitoribosomes follow different biogenesis routes by forming protein-only submodules during early assembly, which are RNA-independent[5]. Later assembly steps ensure the proper folding of critical regions at the subunit interface, such as the peptidyltransferase center (PTC) and the decoding center (DC). These steps depend on biogenesis factors including RNA modifying enzymes like methyltransferases, helicases, chaperones and a conserved group of GTPases[10]. In particular, PTC

[1]Department of Molecular Biology, University Medical Center Göttingen, Göttingen, Germany. [2]Cluster of Excellence "Multiscale Bioimaging: from Molecular Machines to Networks of Excitable Cells" (MBExC), University of Göttingen, Göttingen, Germany. [3]Department of Cellular Biochemistry, University Medical Center Göttingen, Göttingen, Germany. [4]Research Group Structure and Function of Molecular Machines, Max Planck Institute for Multidisciplinary Sciences, Göttingen, Germany. [5]Department of Molecular Biology, Max Planck Institute for Multidisciplinary Sciences, Göttingen, Germany. [6]Max Planck Institute for Multidisciplinary Sciences, Göttingen, Germany. [7]Fraunhofer Institute for Translational Medicine and Pharmacology ITMP, Translational Neuroinflammation and Automated Microscopy, Göttingen, Germany. [8]Göttingen Center for Molecular Biosciences, University of Göttingen, Göttingen, Germany. [9]These authors contributed equally: Marleen Heinrichs, Anna Franziska Finke. ✉e-mail: hauke.hillen@med.uni-goettingen.de; ricarda.richter@med.uni-goettingen.de

folding depends on MRM2, MTERF4-NSUN4 and the GTPases GTPBP5, −6, −7, and −10, as recently revealed by high-resolution cryo-EM structures[11–18]. Similarly, mtSSU maturation requires MCAT, mtRBFA, the methyltransferases METTL17, -15 and TFB1M, and the GTPases ERAL1 and MTG3[19,20].

Loss of MTG3 (also called NOA1 or C4ORF14) causes mitochondrial translation deficiency in different model systems; embryonic lethality in mice and reduced cell viability in isolated MEFs suggest a crucial role of MTG3 in mitoribosome assembly, however the molecular consequences of MTG3 ablation have remained to be addressed[21,22]. Initial studies have shown that MTG3 interacts specifically with the mtSSU and not with the mtLSU or 55S, indicating a function during mtSSU assembly, similar to its bacterial homolog YqeH[10,22]. A previous study has shown that multiple mutations within the GTPase domain impair the ability of MTG3 to bind mtSSU particles, thus indicating the requirement of GTP-binding or -hydrolysis for mtSSU biogenesis[22]. Recent structural data have revealed its binding site at the subunit interface of the maturing mtSSU and suggest that MTG3 is bound very early in the assembly pathway, when the head is still immature[20]. The release of MTG3 from the mtSSU has been suggested to occur prior to binding of mS38 and mtRBFA[20]. However, the precise role of MTG3 during mtSSU biogenesis and how its release from the maturing subunit is triggered remain unknown.

Here, we combine genetic perturbation with biochemical and structural analyses to elucidate the role of MTG3 during mtSSU biogenesis. We show that loss of MTG3 leads to a substantial decrease in mitochondrial translation due to a disturbed mtSSU assembly. Surprisingly, immunoprecipitation experiments of MTG3-containing mtSSU complexes reveal the co-isolation of translation initiation factors. Cryo-EM structures of MTG3-bound mtSSU particles show that MTG3 remains bound to late maturing particles and even to initiation complexes via its N-terminal domain (NTD), preventing the docking of an rRNA helix (h44) and therefore subunit joining. Taken together, these data suggest that MTG3 may act as a quality control factor that couples late mtSSU maturation with the formation of primed translation initiation complexes.

## Results

### MTG3 is required for mtSSU biogenesis

To study the role of MTG3 in mitoribosome biogenesis, we generated MTG3 knockout cell lines using CRISPR/Cas9 technology with guide RNAs targeting its first exon. Two knockout clones were isolated for which MTG3 is not detectable via western blotting (Fig. 1a). Genomic DNA sequencing shows that both clones contain premature stop codons, leading to truncated and presumably unstable variants of MTG3 (Fig. 1b, Supplementary Fig. 1a). As both clones show a similar behavior, we proceeded with only one of them for further downstream approaches. The loss of MTG3 significantly affects cell growth and is accompanied by rapid acidification of the media even with high-glucose, indicating a mitochondrial dysfunction (Fig. 1c). Indeed, oxygen consumption rate (OCR) is strongly reduced in *Mtg3⁻/⁻* while extracellular acidification rate (ECAR) is elevated (Supplementary Fig. 1b). In agreement with decreased OXPHOS capacity we observed reduced *in gel* activity for respiratory chain complexes I and IV (Supplementary Fig. 1c). To understand the reason of the OXPHOS deficiency, we monitored mitochondrial translation by [³⁵S]Methionine incorporation into newly synthesized mtDNA-encoded proteins, which is strongly impaired, but not completely abolished (Fig. 1d). Mitochondrial protein synthesis is restored upon expression of a FLAG-tagged variant of MTG3 in the knockout background, excluding possible off-target effects in the knockout cell line and confirming that tagged MTG3 is physiologically functional (Fig. 1d). To dissect the underlying basis of this translation defect, we analyzed the protein steady state levels of multiple nucleus-encoded components of the mitochondrial gene expression machinery, in particular ribosomal proteins, assembly- and

translation factors (Fig. 1e, f). Interestingly, we observe a differential reduction in multiple MRPs of the mtSSU. The mt-rRNA-dependent MRPs uS14m and uS15m and the late binding protein mS37 are drastically decreased to 20–30%, whereas other proteins remain more stable, indicating a role of MTG3 in late maturation steps. In contrast, assembly factors such as the methyltransferases TFB1M and NSUN4 or the GTPase ERAL1 are slightly elevated. A notable exception is mtRBFA, which is slightly reduced. Components of the mtLSU are not affected or slightly increased, consistent with a role of MTG3 in mtSSU- but not mtLSU biogenesis. In agreement with the reduced protein steady state levels of mtSSU MRPs, we observe a significant reduction of 12S mt-rRNA steady state level to 40%, whereas the 16S mt-rRNA remains stable (Fig. 1g, h). The level of mRNA encoding for COX1 (*MT-CO1*) does not differ, suggesting that the observed defect in mitochondrial translation is not caused by a decrease in mitochondrial transcripts. To investigate the consequences of MTG3 ablation on mtSSU biogenesis in more detail, we separated mitoribosomal complexes from both wildtype and *Mtg3⁻/⁻* cells by sucrose density gradient centrifugation (Fig. 1i). MTG3 is only detectable in less dense (2-5) and in mtSSU-corresponding fractions (6/7) in the wildtype sample, consistent with a role of MTG3 during mtSSU biogenesis as previously suggested[22]. In agreement with the reduction in de novo mitochondrial translation, we observe a strong decrease in assembled 55S mitoribosomes in fraction 11 when comparing *Mtg3⁻/⁻* to the wildtype sample. The mtSSU in fractions 6/7 is also drastically reduced, as monitored with individual mtSSU components. However, mtSSU proteins such as mS40 or mS27 are unaffected in less dense fractions 2 to 4, indicating that intermediate complexes of the mtSSU can be stably formed independently of MTG3. Interestingly, assembly factors ERAL1 and TFB1M remain detectable or are slightly increased in mtSSU-corresponding fractions 6/7 although the overall level of mtSSU is decreased, indicating a stalling in the mtSSU assembly pathway. In contrast, mtRBFA is significantly decreased in fraction 6/7, suggesting that MTG3 action is required upstream of mtRBFA. Components of the mtLSU accumulate in fractions 8/9, as they cannot proceed to form functional 55S mitoribosomes due to the absence of sufficient matured mtSSU in *Mtg3⁻/⁻*. In conclusion, MTG3 ablation affects mtSSU biogenesis, leading to a reduced pool of translationally active mitoribosomes.

### MTG3 associates with multiple mtSSU biogenesis states

To reveal the precise role of MTG3 during mtSSU biogenesis, we inducibly expressed a FLAG tagged variant of MTG3 in HEK293 cells and purified endogenous MTG3-containing complexes via MTG3^FLAG-co-immunoprecipitation (Fig. 2a). Under the chosen conditions, all tested MRPs of the mtSSU co-purified via MTG3 while components of the mtLSU are not detectable, validating the assumption that MTG3 only associates with mtSSU complexes, but not with the mtLSU or the 55S mitoribosome (Fig. 2a). ERAL1 and TFB1M, but not NSUN4 co-purify with MTG3-containing mtSSU particles, suggesting that MTG3 is part of an assembly intermediate with these and other factors, consistent with previous reports[20]. A reciprocal experiment using ERAL1^FLAG in an *Eral1⁻/⁻* background as a bait likewise results in the co-isolation of MTG3 and TFB1M, thus supporting this hypothesis (Supplementary Fig. 1e). In contrast to previous observations, however, MTG3 also associates with factors characteristic of late mtSSU biogenesis steps. For example, mtRBFA, one of the final mtSSU assembly factors[19], can be also co-purified with MTG3. Likewise, mS37, which is the last MRP that joins the maturing mtSSU[19], is also detectable, although to a slightly lesser extent compared to other mtSSU constituents. This suggests that MTG3 remains bound to very late assembly states of the mtSSU.

To further investigate the role of MTG3 during late maturation of the mtSSU, we analyzed the endogenous MTG3-containing ribosomal complexes isolated via co-immunoprecipitation with MTG3^FLAG as the bait by single-particle cryo-electron microscopy (cryo-EM). After

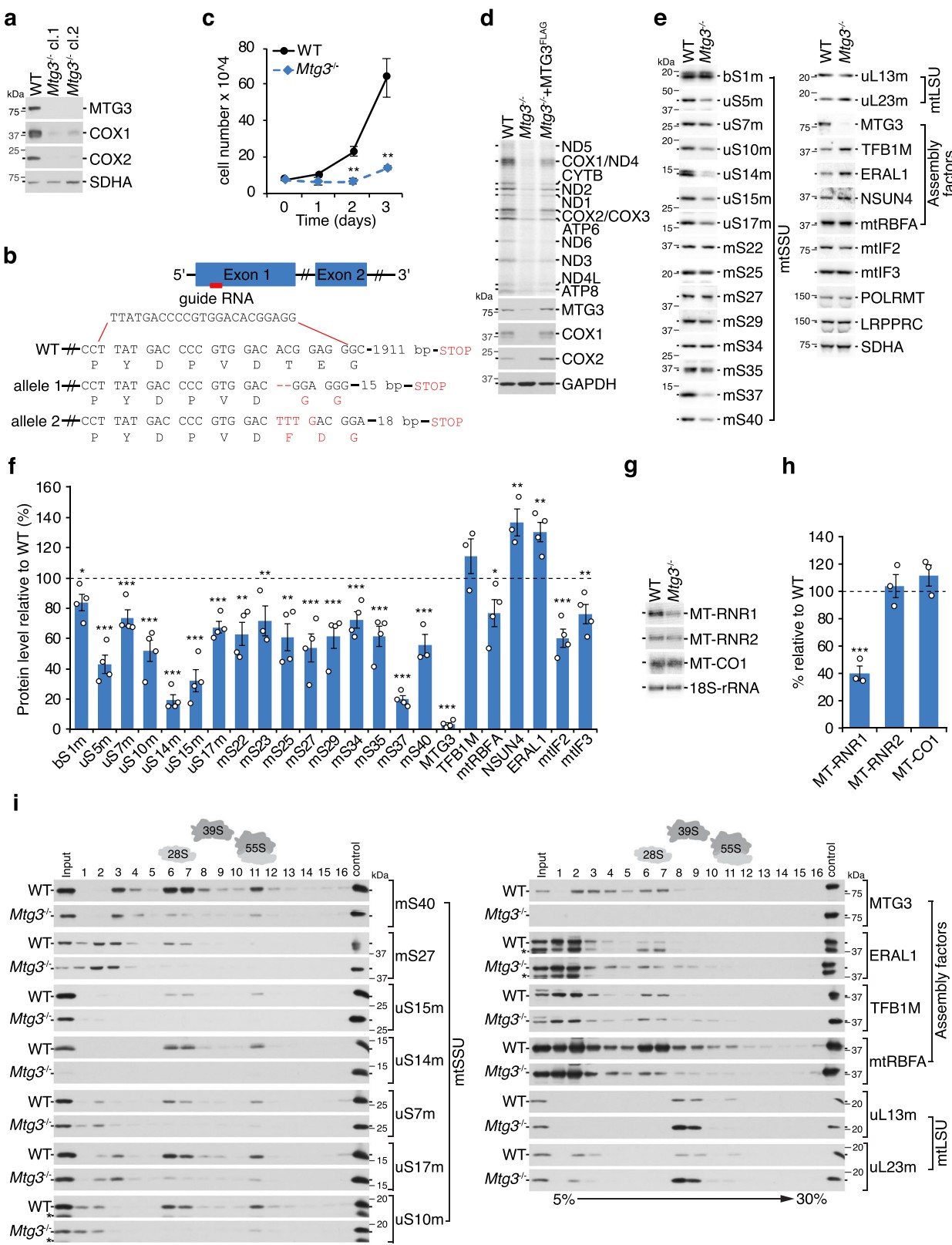

extensive particle classification, we obtained reconstructions for three distinct mtSSU assembly intermediates with immaturely folded 12S rRNA (States A-C) at overall resolutions ranging from 3.1 to 3.4 Å (Fig. 2b, c Supplementary Figs. 2, 3, Supplementary Table 1).

State A represents the earliest intermediate exhibiting the highest degree of immaturely folded rRNA helices in the 12S rRNA (Fig. 2b, d).

Compared to the mature mtSSU[2], we observe three additional densities which could be unambiguously assigned as assembly factors MTG3, TFB1M and mtRBFA, together constituting a combination of assembly factors which has not been described to date. Consistent with recently reported structural data[20], MTG3 consists of a C-terminal globular GTPase domain, which resides on the face of the 12S rRNA,

**Fig. 1 | Loss of MTG3 leads to disturbed small subunit assembly. a** Confirmation of MTG3 knock out in two cell lines generated using CRISPR/Cas9 technology. Isolated mitochondria (10 μg) from wildtype and *Mtg3⁻/⁻* cell lines (cl.1 and cl.2) were analyzed by western blotting with antibodies as indicated. Similar results were obtained in *n* ≥ 3 biologically independent experiments. **b** Schematic representation of the genomic locus of the generated *Mtg3⁻/⁻* cell line (cl.1) in comparison to the wild type sequence. The guide RNA targets exon 1 of the *MTG3* gene, which encodes for a 648 aa protein. A two bp deletion and a four bp insertion in the two alleles of the *Mtg3⁻/⁻* cl.1 lead to premature stop codons and truncated proteins (65 aa). **c** Ablation of MTG3 reduces growth rate. Equal amounts of wild type and *Mtg3⁻/⁻* cells were seeded in three biologically independent experiments on day 0 and counted at the indicated time points (*n* = 3; mean ± SEM). Significance was calculated by two-sample one-tailed Student's *t*-test and defined as **p ≤ 0.01. **d** Translation of mtDNA-encoded proteins is disturbed upon loss of MTG3. Mitochondrial translation in wild type, *Mtg3⁻/⁻* and *Mtg3⁻/⁻* cells inducibly expressing MTG3^FLAG was analyzed via [³⁵S]Methionine de novo incorporation and subsequently visualized via autoradiography and with indicated antibodies. The signal in *Mtg3⁻/⁻* using MTG3 antibody represents unspecific binding of the antibody in whole cell lysates as we confirmed several times the loss of MTG3 in isolated mitochondria. Similar results were obtained in *n* ≥ 3 biologically independent experiments. **e**, **f** MTG3 loss leads to reduced mtSSU MRP level. Steady state analysis of MRPs, assembly factors, and translation-related proteins in the *Mtg3⁻/⁻* cells in comparison to wild type cells. Isolated mitochondria were analyzed via western blotting with indicated antibodies (**e**) and protein levels in *Mtg3⁻/⁻* were quantified relative to wild type control (**f**). SDHA was used as a loading control. Statistical analysis was performed as two-sample one-tailed Student's *t*-test with *n* ≥ 3 biologically independent samples shown as mean ± SEM (individual data points are shown as circles). Significance was defined as *p ≤ 0.05, **p ≤ 0.01, ***p ≤ 0.001. **g**, **h** Effect of MTG3 loss on rRNA and mRNA stability. **g** RNA isolated from *Mtg3⁻/⁻* and wild type cells was subjected to northern blotting using indicated probes (*MT-RNR1*: 12S rRNA; *MT-RNR2*: 16S rRNA; *MT-CO1*: mRNA encoding for COX1). 18S-rRNA was used as loading control. **h** Quantification of RNA signals in *Mtg3⁻/⁻* from (**g**) relative to wild type signals. Statistical analysis was performed as two-sample one-tailed Student's *t*-test with *n* = 3 biologically independent samples shown as mean ± SEM (individual data points are shown as circles). Significance was defined as ***p ≤ 0.001. **i** mtSSU and monosome levels are severely reduced in *Mtg3⁻/⁻* cells. Isolated mitoplasts (500 μg) were lysed and subjected to sucrose density gradient centrifugation. Fractions (1-16) were collected and analyzed via western blotting with antibodies against MRPs and assembly factors as indicated. Input = 10% of total. Similar results were obtained in *n* ≥ 3 biologically independent experiments.

and a N-terminal domain (NTD), which occupies the binding site of an extended lasso structure in the mature h44 (1501-1549)[19,20] (Fig. 2c, e). MTG3 binding to the mtSSU thus prevents folding of h44 into its mature state via its globular domain as well as its NTD (Supplementary Fig. 4a). TFB1M binds next to MTG3 and assists in maturation of the DC by carrying out a di-methylation reaction in h45[23]. The rRNA adopts a premature conformation in this state, with a completely disordered h44 (1481-1572), partially disordered h45, and a misplaced h24 in the DC due to interactions with TFB1M and mtRBFA (Fig. 2b, d).

State B and C represent later assembly intermediates in which the 12S rRNA is partially matured (Fig. 2f–i). They both do not show densities for the MTG3 globular domain or TFB1M, but contain mtRBFA and an additional density, which corresponds to the methyltransferase METTL15 (Fig. 2c). mtRBFA has previously been shown to adopt two different states on the mtSSU, termed "in" and "out"[19]. While states A and B contain mtRBFA in the "in" conformation, it adopts the "out" conformation in state C, which has been reported to be a hallmark of very late-stage assembly steps[19]. METTL15 has been shown to interact with h24 and h44 to methylate residue C1486 in h44[24] (Supplementary Fig. 4b). Consistent with this, h24 and the region of h44 containing C1486 are moved towards METTL15 compared to their locations in the mature mtSSU (Fig. 2f, h, Supplementary Fig. 4b). In both state B and C, h18 and h27 adopt a mature conformation and mS38 is present. Although we do not observe density for the MTG3 globular domain in these states, the density is consistent with the NTD of MTG3 remaining associated with the mtSSU in the h44 binding site. Consistent with this, the trajectory of h44 faces away from the foot of the mtSSU and the lasso region (1501-1549) appears absent (Fig. 2c, g, i, Supplementary Fig. 4a). These observations suggest that MTG3 may remain associated to the mtSSU even during late assembly steps via its NTD, thereby preventing docking of h44 and in turn binding of the mtLSU.

**Molecular function of MTG3 during mtSSU maturation**

A recent study provided structural snapshots of MTG3 bound to the mtSSU together with the assembly factors TFB1M, MCAT, METTL17, and ERAL1[20] (Fig. 3a, b). While TFB1M and MTG3 facilitate the maturation of h27, h44, and h45 in the body, MCAT and METTL17 are thought to mature the rRNA in the head. ERAL1 is located at the interface between head and body, binding the 3'end of the 12S mt-rRNA and recruiting the bS21m−uS11m−h23 module. Recruitment of mtRBFA and docking of the bS21m-uS11m-h23 module causes platform compaction and was suggested to occur after MTG3 dissociation and subsequent h44 docking. Recruitment of this module further induces di-methylation of two residues in the hairpin loop in h45 (A1583/A1584) by TFB1M.

The three states we observe differ from previously reported states both in their composition and their rRNA maturation states (Fig. 3a). In state A, the head is already fully matured but METTL17 and MCAT are not present (Fig. 3b). In addition, ERAL1 is replaced by mtRBFA^(in) and the bS21m−uS11m−h23 module, causing a rotation of the head by 25 Å (Fig. 3b). This also moves the NTD of TFB1M away from the bS21m-uS11m-h23 module to a location that differs from a previously reported late assembly intermediate containing mtRBFA^(in) and TFB1M[19] (Fig. 3a, Supplementary Fig. 5a). In addition, we observe a density that appears to correspond to a flexible helix at the N-terminus of TFB1M (Fig. 3c). Close to this region, METTL17 was previously observed to form contacts with h44 together with the flexible helix of TFB1M (Supplementary Fig. 5c). In contrast, METTL17 is absent in state A and TFB1M may instead form contacts with the partially disordered h18 (Fig. 3c, Supplementary Fig. 5c). Within the rRNA, h44 is disordered to a larger extent than previously observed and does not form any contacts with TFB1M or h45 (Supplementary Fig. 5b). In addition, h18 in the shoulder region is partially disordered and oriented towards TFB1M[19,20] (Fig. 3c, Supplementary Fig. 5c), and h27 is in its mature conformation, although it has been predicted that this would lead to clashes with MTG3 and TFB1M (Supplementary Fig. 5d). This is possible because both assembly factors adopt distinct arrangements in state A compared to previous structures. In the observed conformation, MTG3 would clash with mS38, which locks h27 and h44 in their final conformation in the mature mtSSU, but mS38 is not present in our state A and replaced by interactions between TFB1M, MTG3, and the rRNA helices. Finally, h27 is in its mature conformation in state A (Supplementary Fig. 5d), which is a prerequisite for bS21m-uS11m-h23-module recruitment, platform compaction and TFB1M-NTD re-orientation. This is in contrast to the previous structure of a MTG3- and TFB1M-bound intermediate, in which h27 adopts a more immature conformation (Supplementary Fig. 5d) and which suggested that platform compaction and TFB1M catalytic activity is precluded by MTG3-bound h27 until h44 docking has occurred. Taken together, state A differs substantially from previously reported MTG3-containing mtSSU intermediates as the rRNA fold in the head and platform more closely resembles later stages of mtSSU maturation[19], even though h44 docking and h18 maturation has not yet occurred.

We also observed differences in MTG3 compared to previously reported data. The NTD of MTG3 is composed of the mS27-interacting peptide (residues: 69-79) which anchors the NTD to the mitoribosome followed by a long α-helix sitting on the base of the platform (residues:

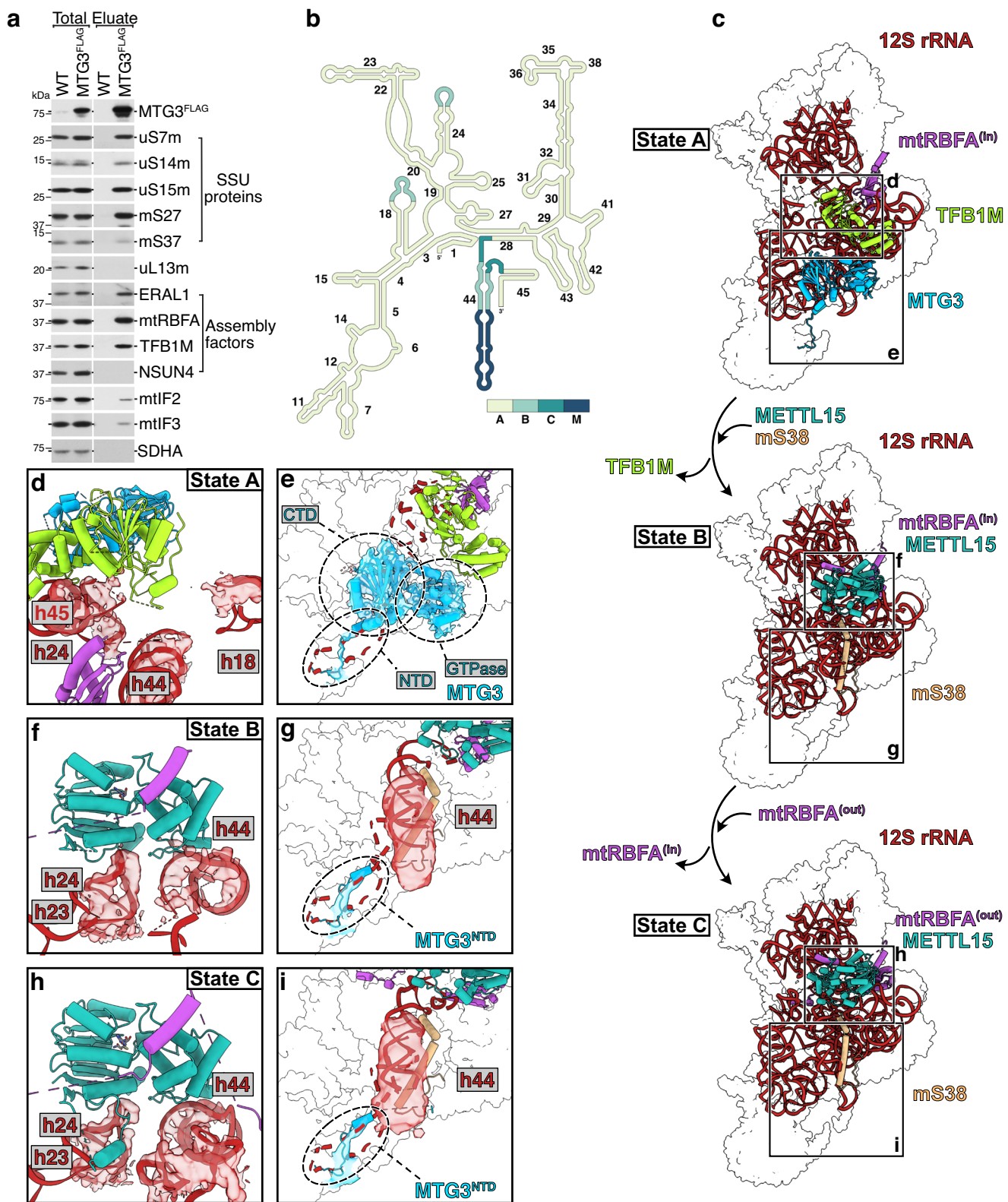

87-115) (Supplementary Fig. 5e). The NTD is connected via a linker to a helical insertion (residues: 179-195) at the GTPase domain of MTG3. This helical insertion was previously observed to partially occupy the empty GTP-binding site, preventing switch I loop from entering a GTP-binding competent conformation. Based on the bacterial homolog YqeH, it has been suggested that the helical insertion and switch I loop rearrange during the GTPase cycle, which triggers h27 and h44 maturation[21,25]. In state A, the long α-helix and the helical insertion of

MTG3 are disordered and their previously observed position is occupied by uS15m (Fig. 3d, Supplementary Fig. 5e). In contrast to previous MTG3-bound structures, the N-terminal helix of uS15m is ordered and occupies the position of the long α-helix (Fig. 3d). In this conformation of MTG3, the nucleotide-binding site is accessible and the switch I loop may adopt a conformation competent for GTP-binding. In agreement with this, we observe a density in the nucleotide binding pocket which would be consistent with a bound nucleotide (Fig. 3d, Supplementary

**Fig. 2 | Structures of mtSSU assembly intermediates co-isolated via MTG3^FLAG.**
**a** MTG3^FLAG co-isolates mtSSU MRPs and several mtSSU assembly factors. FLAG-immunoprecipitation was performed with lysed mitochondria from HEK293 wild type (WT) cells and a stable HEK293 cell line inducibly expressing MTG3^FLAG, and subsequently analyzed via western blotting with indicated antibodies (total = 3%, eluate = 100%). Similar results were obtained in n ≥ 3 biologically independent experiments. **b** Schematic depiction of the mature 12S rRNA secondary structure. Regions are depicted by their level of maturation in each state, with M corresponding to the mature mtSSU (PDB: 3J9M[2];). **c** Cryo-EM structures of the MTG3-TFB1M-mtRBFA^(in)-bound small mitoribosomal subunit (mtSSU) intermediate (state A), METTL15-mtRBFA^(in)-bound mtSSU (state B), and METTL15-mtRBFA^(out)-bound mtSSU (state C). The 12S rRNA (red) and indicated biogenesis factors are shown as cartoon and the remaining mitoribosomal proteins (MRPs) are indicated as white transparent surface. MTG3: light blue, TFB1M: lime green, mtRBFA: indigo, METTL15: turquoise, mS38: beige[2]. Close-up views of the immature decoding center (**d**) and of the foot region (**e**) in state A. Coloring as in (**c**), with cryo-EM densities (from map A3) of immature rRNA helices shown as red surface. Close-up views of the immature decoding center with cryo-EM densities (from maps B-C3) (**f**, **h**) and of the foot region (**g**, **i**) in state B and C, respectively. (**e**, **g**, **i**) Close-up views of the foot region in each state, showing the densities for MTG3 (state A, map A3), or MTG3^NTD and h44 with altered trajectory (state B-C, 15 Å low-pass filtered maps B1 and C1). Mature h44 is depicted by red dashed lines and would clash with MTG3 and MTG3^NTD.

Fig. 5f)[25]. However, we refrained from modeling a ligand due to the limited resolution of the map in this region. Taken together, these data indicate that MTG3 can adopt multiple conformations while bound to the mtSSU and that its GTP-bound conformation is compatible with mature folding of h27 and binding of uS15m. This links MTG3 GTPase function to docking of its own NTD into the foot region and re-organization of rRNA content, but also to re-orientation of an MRP.

## METTL15 induces a conformational switch of mtRBFA

The structural data also provide insights into the role of METTL15 during mtSSU biogenesis. METTL15 interacts with h44 to methylate m⁴C1486, but this residue is ~40 Å away from its catalytic center in its mature conformation in state C (Supplementary Fig. 4b), resembling a previously described post-catalytic state[19]. In contrast, while the overall fold of the rRNA in states B and C resembles previously observed conformations, the segment containing m⁴C1486 appears less well ordered in state B, suggesting that it may represent an intermediate state previously not observed (Fig. 3e-f, Supplementary Fig. 4b).

Previous studies have suggested that METTL15 plays an important structural role during mtSSU biogenesis in addition to its catalytic activity as a methyltransferase[19]. Consistent with this previous report, we observe METTL15 together with mtRBFA in its "out" conformation (mtRBFA^(out)) in state C (METTL15_C). By contrast, state B contains mtRBFA in its "in" conformation together with METTL15_B, a combination that has not been previously described (Fig. 3a, e). This is possible because a mitochondria-specific extension of METTL15, which would clash with mtRBFA^(in), is disordered in METTL15_B (Fig. 3g, Supplementary Fig. 4b). In previous mtRBFA^(in) containing structures, its C-terminal extension (mtRBFA-CTE) was observed to be disordered. By contrast, we observe a density at the interface to METTL15_B, which is consistent with the presence of the mtRBFA-CTE (Fig. 3g). This suggests that in state B, the mitochondrial specific extension of METTL15 is disordered, but folds upon m⁴C1486 methylation and maturation of this part of h44 upon transition to state C, thereby inducing a conformational switch of mtRBFA from "in" to "out". The mtRBFA^(in) conformation has been associated with earlier mtSSU intermediates than mtRBFA^(out), and the transition from "in" to "out" was suggested to cause the head to rotate closer to its mature position, constituting a checkpoint in the mtSSU assembly pathway[19]. Our structural data suggest how conformational changes of METTL15 and mtRBFA may be coupled, and thus provide a molecular rationale for this checkpoint.

In all previously reported METTL15 and mtRBFA containing states[19,20] the lasso of h44 adopts its mature conformation. It has thus been suggested that these two factors are hallmarks of the final steps of mtSSU maturation, and are released by mS37 as well as the initiation factor mtIF3 during the transition from assembly to initiation[19]. However, state B and C both contain a disordered h44 lasso, which is not yet docked into the foot region. Instead, its location is occupied by the MTG3^NTD (Fig. 2g, i, Supplementary Fig. 4a). This suggests that docking of h44 and MTG3 dissociation is not necessarily required for late stage mtSSU assembly steps, and that initiation complex assembly could potentially commence already prior to completed mtSSU maturation.

## MTG3 co-purifies with primed initiation complexes

Consistent with a potential link between mtSSU assembly and initiation complex formation, we noticed accumulation of mtSSU initiation factors, namely mtIF2 and mtIF3, together with mtSSU proteins during immunoprecipitation via FLAG-tagged MTG3 (Fig. 2a). Reciprocally, using FLAG-tagged mtIF3 to isolate native mtSSU complexes accumulated mtIF2 as well as the assembly factors mtRBFA and MTG3 (Fig. 4a). Thus, our data suggest that IC intermediates co-purify with the assembly factor MTG3 and vice versa. In agreement with this, particle classifications showed that a substantial proportion of mtSSU particles in our cryo-EM dataset indeed had translation initiation factors bound (Supplementary Fig. 2). This enabled us to resolve four distinct (pre-) translation initiation complexes ((P)-IC) (states D-G) at overall resolutions ranging from 3.1 to 3.6 Å (Fig. 4b, Supplementary Figs. 2, 3, Supplementary Table 2).

State D resembles a previously described structure of the PIC with mtIF3 bound[19]. State E additionally contains mtIF2, thus resembling another recently reported PIC[26]. State F represents a so-far undescribed PIC state that contains only mtIF2. State G contains mtIF2 and fMet-tRNA^Met[19]. In this state, we observe clear density for the codon-anticodon interaction, consistent with a published IC state primed for translation initiation (Fig. 4c)[19]. In all four (P-)IC states, most of the mt-rRNA is matured and mtRBFA is replaced by mS37, consistent with previous data (Fig. 4b, Supplementary Table 3). However, h44 remains immature in all PICs, similar to the assembly states B and C, with the trajectory of the lasso region protruding from the complex instead of docking into the foot region. In addition, the foot region shows density which is most consistent with the presence of the MTG3^NTD (Fig. 4d–g, Supplementary Table 3). This suggests that h44 maturation is not required to start IC formation and that translation initiation factors can be recruited before mtSSU maturation is finalized. However, the association of MTG3 during first steps of IC formation prevents docking of h44 and thus mtLSU recruitment to the mtSSU.

## The N-terminal region is essential for MTG3 functionality

The N-terminal region of MTG3 displaces h44 from its mature conformation and is highly conserved across vertebrates (Supplementary Fig. 6a)[20]. Our data indicate that it may remain stably bound to the mtSSU, even when mS38 binds and the globular domain of MTG3 is displaced (Fig. 2g, i, Fig. 4d–g, Supplementary Table 3). To further dissect the impact of this domain on MTG3 function, we expressed a FLAG-tagged mutant variant of MTG3 lacking the highly conserved 10 aa (69-83) which interact with mS27 and displace h44 (ΔN-MTG3^FLAG) in the *Mtg3^-/-* cell line (Supplementary Fig. 6a). Deletion of these 10 aa does not interfere with the import of ΔN-MTG3 into mitochondria, as the upstream N-terminal pre-sequence remains preserved and ΔN-MTG3^FLAG is comparably detectable in isolated mitochondria like the wildtype variant of MTG3^FLAG.

| STATE | Assembly factors | | | | | | | Matured module | | | | | | |
|---|---|---|---|---|---|---|---|---|---|---|---|---|---|---|
| | MTG3 | TFB1M | mtRBFA | METTL15 | MCAT | ERAL1 | METTL17 | Head | h44 | h45 | h27 | h24 | h23 | h18 |
| STATE A | + | + | + | - | - | - | - | + | - | - | + | - | + | - |
| 8CSP | + | + | - | - | + | + | + | - | - | - | - | - | - | + |
| 7PNT | - | + | + | - | - | - | - | + | - | - | + | - | + | + |
| State B | (+) | - | + | + | - | - | - | + | - | - | + | - | + | + |
| State C | (+) | - | + | + | - | - | - | + | - | - | + | - | + | + |
| 7PNX | - | - | + | + | - | - | - | + | (+) | - | + | - | + | + |
| 3J9M | - | - | - | - | - | - | - | + | + | + | + | + | + | + |

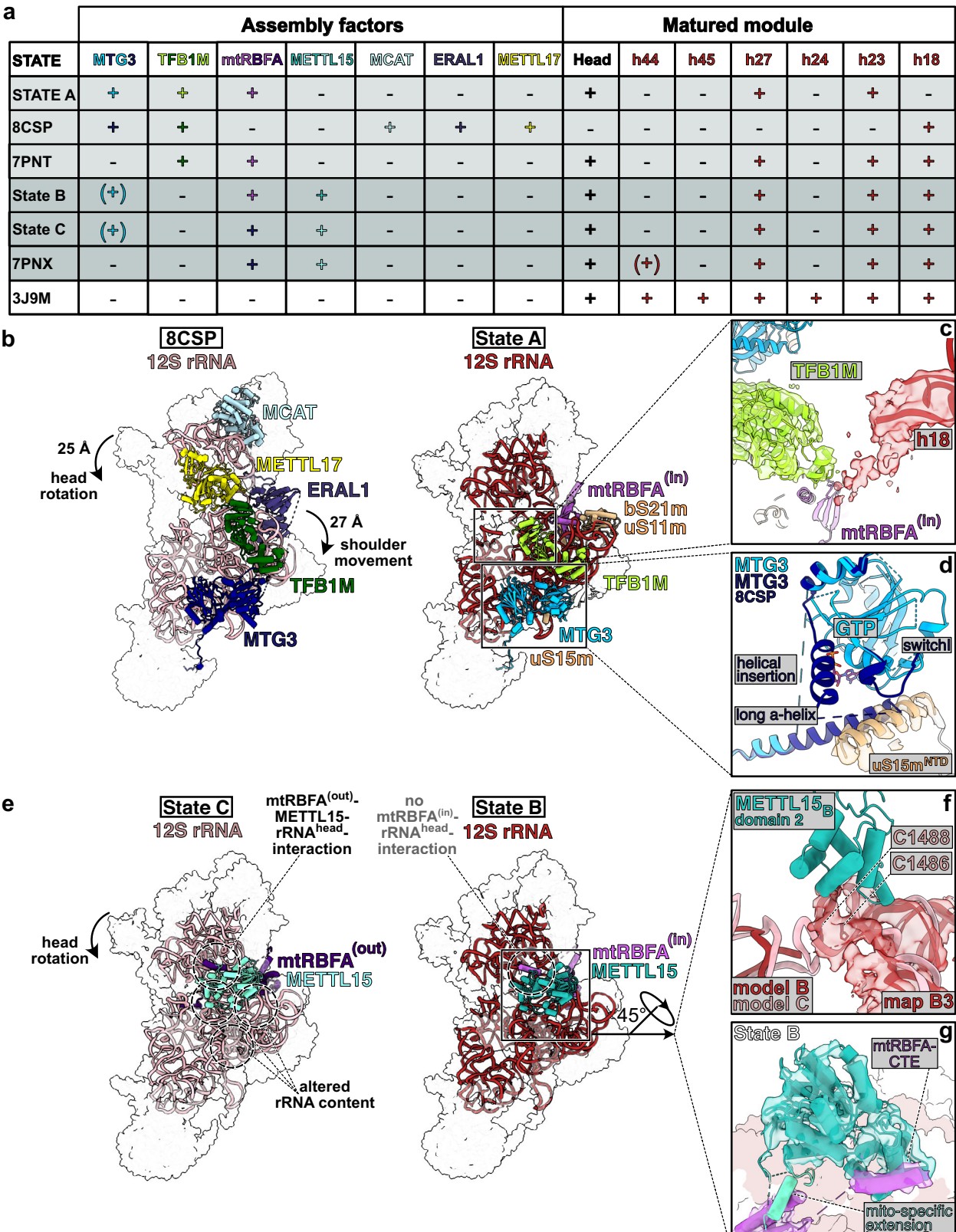

To dissect the role of this N-terminal helix, we monitored mitochondrial translation by [³⁵S]Methionine de novo synthesis in *Mtg3*^−/− expressing MTG3^FLAG wildtype or ΔN-MTG3^FLAG mutant. Although mitochondrial translation seems to be slightly increased upon ΔN-MTG3^FLAG expression in comparison to the full knockout, in contrast to full-length MTG3, the ΔN-MTG3^FLAG can only partially restore mitochondrial translation (Fig. 5a). Interestingly, we did not observe a uniform restored translation pattern. While translation of e.g., ATP6 and ATP8 seems to be completely rescued in the mutant cell line, the levels of newly synthesized COX1 and corresponding protein steady state are only marginally increased in comparison to the full knockout, but by far not restored as by expressing MTG3^FLAG. This indicates that

**Fig. 3 | Structural comparison of assembly intermediates. a** Table of six described assembly intermediates each comprising at least one of the assembly factors MTG3, TFB1M, or mtRBFA, as well as the mature mtSSU (PDB: 3J9M[2];). Maturation of 12S rRNA modules is shown for each state (matured: "+", unmatured: "−"). **b** The previously described assembly state (PDB 8CSP[20]) (left) as well as state A (this study) (right) are shown side-by-side, with additional modules highlighted (right). The 12S rRNA and indicated biogenesis factors are shown as cartoon and the remaining mitoribosomal proteins (MRPs) are indicated as white transparent surface. Depiction as follows (state A vs PDB 8CSP[20]): 12S rRNA: red vs peach, MTG3: light blue vs dark blue, TFB1M: lime green vs dark green, mtRBFA: indigo, bS21m, uS21m, uS15m: beige, METTL17: yellow, MCAT: light blue, ERAL1: dark purple.

**c** Zoomed-in view of h18 and TFB1M in state A **d** Zoomed-in view of MTG3 (light blue) and uS15m (beige) in state A vs additional ordered regions of MTG3 in a previous state (PDB 8CSP[20]) (**c, d**) Densities from map A3 (**c**) and A2 (**d**) are depicted as transparent surfaces. **e** Side-by-side view of state B (right) and C (left) with coloring as followed: 12S rRNA: red vs peach, METTL15: dark vs light turquoise, mtRBFA: indigo vs dark violet. **f** Different routes of the rRNA content in state B vs state C. The model of state C is shown superimposed to the model of state B, with 12S rRNA density from state B (map B3) depicted transparent. **g** Close-up view of METTL15-mtRBFA[(in)] interaction in state B superimposed with METTL15 from state C and depicted mito-specific extension from of METTL15. The 12S rRNA and MRPs are indicated as transparent surface.

the N-terminal region of MTG3 is essential for its full functionality. This is further supported by the fact that the mutated ΔN-MTG3[FLAG] cannot co-immunoisolate any mtSSU proteins, assembly factors or translation initiation factors, indicating that the N-terminal region is responsible for stable binding of MTG3 to the mtSSU (Fig. 5b). However, as indicated by the partially restored translation phenotype, translationally active mitoribosomes must be formed in Mtg3[−/−]+ΔN-MTG3[FLAG] cells.

Interestingly, sucrose gradient analysis revealed similar levels of 55S mitoribosomes in fraction 11 comparing Mtg3[−/−]+MTG3[FLAG] and Mtg3[−/−]+ΔN-MTG3[FLAG], but different levels of assembled mtSSU in fractions 6/7 (Fig. 5c, d). While expressing MTG3[FLAG] in Mtg3[−/−] completely rescues the mtSSU assembly defect and leads to comparable levels of assembled mtSSU as the wildtype control, ΔN-MTG3[FLAG] appears incapable of restoring the pool of assembled mtSSU. In contrast, mtSSU assembly intermediates in the lower dense fractions seem to accumulate in the ΔN-MTG3[FLAG]-expressing cell line. The relative amount of assembly factors such as ERAL1, TFB1M, and mtRBFA co-migrating with the mtSSU is increased, considering the decreased amount of mtSSUs in fraction 6/7 in the ΔN-MTG3[FLAG] cell line. This suggests that the majority of mtSSU in ΔN-MTG3[FLAG]-expressing cells represent immature biogenesis intermediates. Nevertheless, some assembled mtSSUs seem to be able to bind the mtLSU and are thus directly transferred to the monosome pool in fraction 11, but the mitochondrial translation assay indicates that those particles are not as active as when expressing MTG3[FLAG] wildtype. Thus, the stable binding of MTG3 to the mtSSU during late maturation steps might ensure quality control steps and prevent the association of immature mtSSU to the mtLSU. However, as the mutant variant is not able to stably interact with the mtSSU to prevent subunit joining, some 55S particles might be formed containing defective mtSSU.

To elaborate the nature of these particles further, we asked whether they have mt-mRNA bound and whether MTG3 plays an active role in mt-mRNA loading during translation initiation. Thus, we measured the abundance of mt-mRNAs in fraction 11 relative to the 16S mt-rRNA and did not observe a strong reduction of mitoribosome-associated mt-mRNAs (Fig. 5e, Supplementary Fig. 6b). In fact, some mt-mRNAs, including MTCO1, appear to be elevated in Mtg3[−/−]+ΔN-MTG3[FLAG] compared to Mtg3[−/−]+MTG3[FLAG] cells, which might indicate a compensatory mechanism to counteract the inefficient translation of COX1. Nevertheless, the overall mt-mRNA loading does not seem to be affected, suggesting that the deletion in the NTD of MTG3 and subsequently the potential loss of bound MTG3 in the late-maturation steps does not affect the mt-mRNA binding to the mtSSU.

In order to study the role of the GTPase domain of MTG3 during mtSSU maturation similar mutational analyses were performed. A mutant variant of MTG3 with a glycine to proline substitution (G499P) in the G3 motif (switch II) was expressed in the Mtg3[−/−] background (MTG3-G499P[FLAG]). The G3 motif interacts with the γ-phosphate of a bound GTP and the G499 was shown to be essential for GTP hydrolysis in other GTPases[10,27,28]. Indeed, the MTG3-G499P[FLAG] cannot restore mitochondrial translation (Supplementary Fig. 6c), indicating that GTPase activity is abolished and mtSSU biogenesis impaired. As the ΔN-MTG3[FLAG] mutant was at least partially able to restore translation

(Fig. 5a), the GTPase domain seems to be more relevant for the role of MTG3 in maturing the mtSSU than the NTD. To further compare and dissect the roles of these two domains, mitoribosome assembly was investigated by sucrose gradient centrifugation with cell lines expressing the mutated variants of MTG3 in comparison to Mtg3[−/−] and Mtg3[−/−]+MTG3[FLAG] rescue (Supplementary Fig. 6d). The MTG3-G499P[FLAG] mutant reveals mtSSU levels in fractions 6/7 and monosomes in fraction 11 comparable to those of Mtg3[−/−], explaining the inability to restore mitochondrial translation. In contrast to the ΔN-MTG3[FLAG] mutant, almost no monosomes are formed in MTG3-G499P[FLAG] mutant. FLAG-immunoprecipitation reveals that the MTG3-G499P[FLAG] can co-isolate mtSSU particles, however it seems that these only resemble early-stage mtSSU intermediates (Supplementary Fig. 6e). While early-binding proteins such as mS27, uS17m and mS40 as well as early assembly factors ERAL1 and TFB1M can be co-isolated, uS15m and the late-binding MRP mS37 as well as the assembly factor mtRBFA are absent in the elution fraction. This indicates that mtSSU biogenesis gets stalled at an early assembly step when the GTPase hydrolysis function of MTG3 is abolished. Taken together, these data suggest that the GTPase activity of MTG3 is essential for mtSSU assembly.

## A molecular model for coupling mitoribosome biogenesis and translation initiation

Our structural and biochemical data suggest that mtSSU assembly and translation initiation are two processes that do not stringently occur sequentially after each other, and allow us to deduce an alternative model of mtSSU biogenesis (Fig. 6). First, the assembly factors TFB1M, MTG3 and mtRBFA[(in)] associate with the mtSSU to facilitate maturation of h44, h45, h44-h45-linker and parts of h27 (Supplementary Table 3). The globular CTD of MTG3 then dissociates, but MTG3 remains associated with the mtSSU via its N-terminal region thereby preventing h44 maturation. This allows the MRP mS38 to be recruited in the next step. The late-step assembly protein METTL15 is then recruited, and its catalytic activity triggers the maturation of h24 and the h44-h45 linker, thereby inducing a conformational switch of mtRBFA from "in" to "out". Upon dissociation of METTL15, mtRBFA[(out)] also dissociates from the mtSSU, which allows mS37 to bind at this position, locking the head in its final rotation towards the body. Although h44 is still immature and MTG3 remains bound via its NTD, translation initiation factors can be recruited to this mtSSU assembly intermediate. From state F onwards, fMet-tRNA[Met] and mt-mRNA associate with the mtSSU generating state G, while h44 remains immature. After priming of the IC by codon-anticodon formation, MTG3 must be released to allow docking and maturation of h44 at the foot and subsequent binding of mtLSU. What triggers the dissociation of MTG3 remains to be addressed. Taken together, our data suggest a mitoribosome assembly and translation initiation pathway in which MTG3 is the last factor to be released from the mtSSU before the complete mitoribosome is formed.

## Discussion
Ribosome biogenesis is an energetically expensive process that requires multiple auxiliary factors to mediate rRNA and protein folding

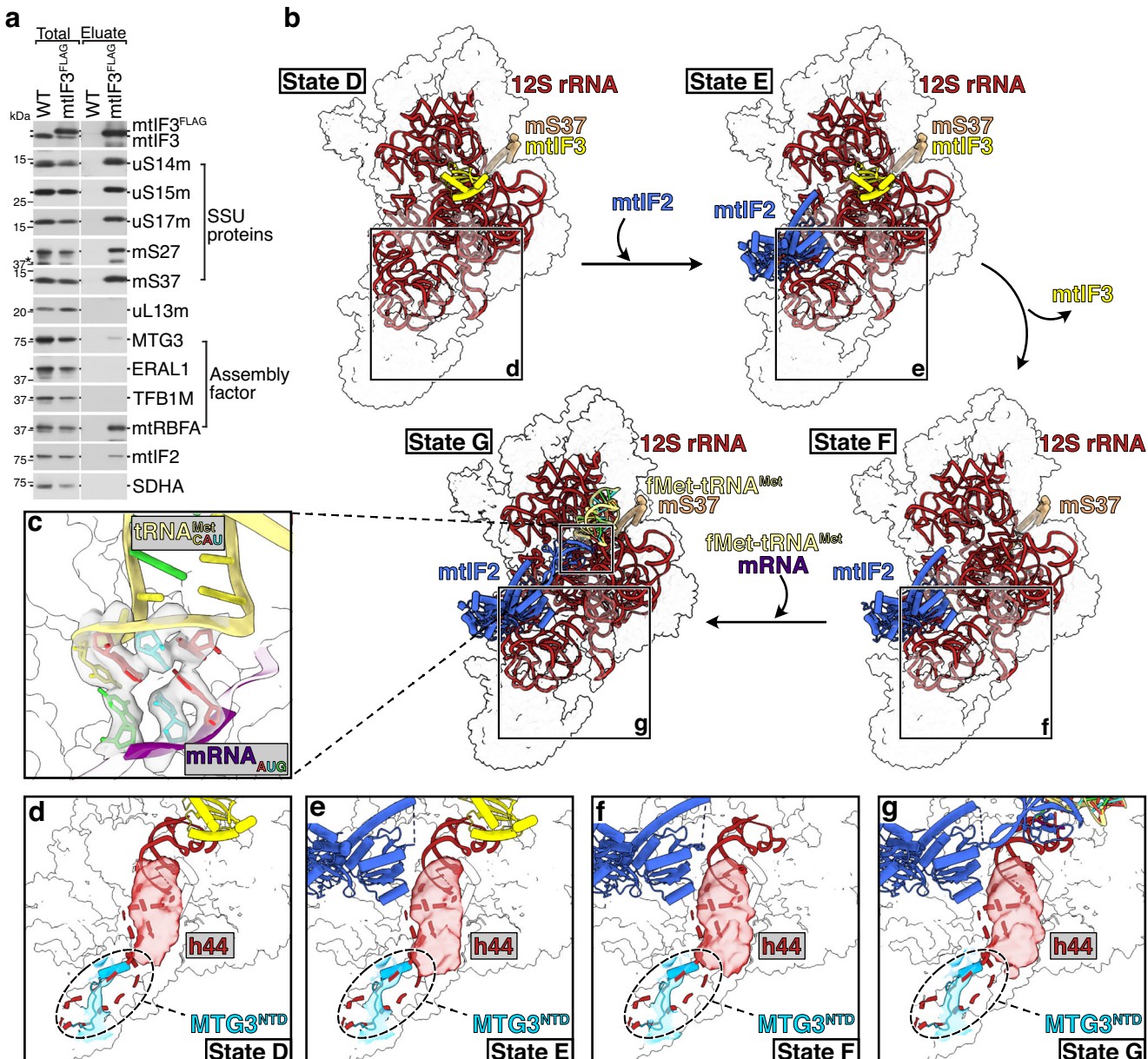

**Fig. 4 | Structures of four mtSSU (pre-) translation initiation states. a** Western blot analysis of mitoribosome complexes co-purified via mtIF3^FLAG. FLAG-immunoprecipitation was performed with lysed mitochondria from wild type cells and cells inducibly expressing mtIF3^FLAG. Samples (total = 3%, eluate = 100%) were subsequently analyzed via western blotting with indicated antibodies. Similar results were obtained in $n \geq 3$ biologically independent experiments. **b** Cryo-EM structures of the mtIF3-bound small mitoribosomal subunit (mtSSU) (state D), mtIF2- and mtIF3-bound mtSSU (state E), mtIF2-bound mtSSU (state F), and mtIF2-, mRNA- and fMet-tRNA^Met-bound mtSSU (state G). The 12S rRNA and indicated initiation factors are shown as cartoon and the remaining mitoribosomal proteins (MRPs) are indicated as white transparent surface. 12S rRNA: red, mtIF3: yellow, mtIF2: blue, fMet-tRNA^Met: light yellow, mS37: beige. **c** Zoomed-in view of the codon-anticodon interaction in state G. Coloring as in (**b**) with mRNA backbone being depicted in purple and nucleotides being highlighted in red (A), blue (U), green (G), and yellow (C). The 12S rRNA and MRPs are indicated as transparent surface. The trajectory of the mRNA is shown by superimposing with a known IC state (7PO2[19];) and depicted as transparent cartoon. **d**–**g** Views of the foot region in each state, showing the densities for MTG3^NTD and for h44 from the 15 Å low-pass filtered maps D1-G1. Mature h44 is depicted by red dashed lines and would clash with MTG^NTD.

and to coordinate the sequential binding of ribosomal proteins. Especially late maturation steps require the assistance of quality control factors which facilitate and monitor the folding of critical regions including the PTC and the DC. Thus, multiple biogenesis factors bind to the intersubunit interface of the mtLSU and mtSSU and thus prevent immature subunit joining. It was assumed for a long time that ribosome biogenesis and translation initiation are separated processes that follow a single stringent route and occur sequentially after each other.

Using a combination of biochemical and structural analysis, we here show that the mtSSU biogenesis factor MTG3 plays a critical role

during late-stage maturation of the mtSSU. We describe several previously unobserved mtSSU assembly intermediates, which provide molecular insights into the role of the biogenesis factors MTG3, METTL15, and mtRBFA. Surprisingly, we were not able to resolve an MTG3-bound mtSSU with associated ERAL1. The reason for this could be that such particles represent a minor fraction of the sample or due to flexibility of the GTPase ERAL1. The challenge of solving GTPases co-purified with assembling mitoribosomal subunits is a common phenomenon in the field[14]. Nevertheless, together with previous data, these structural snapshots enable us to propose a unified model for

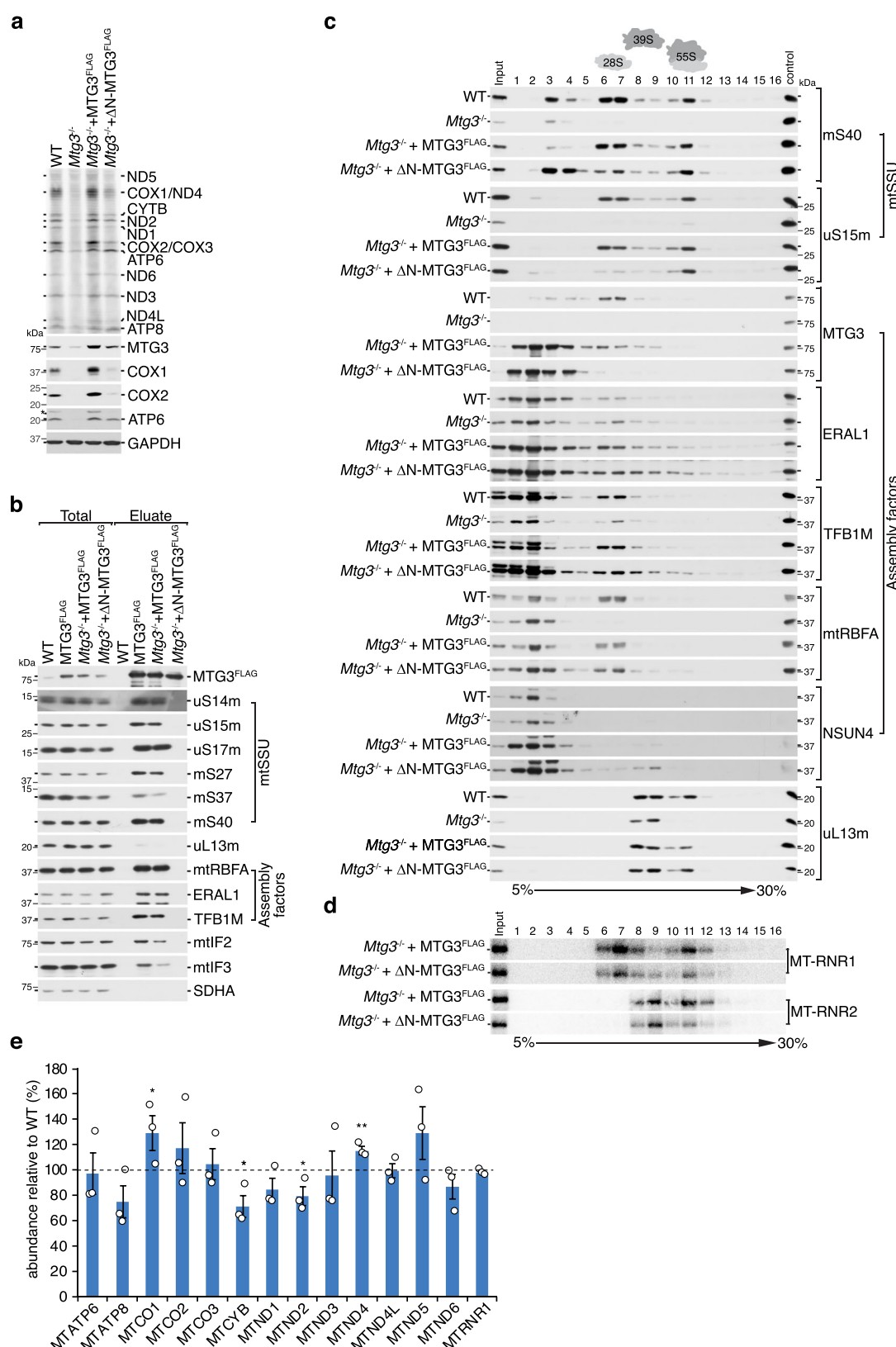

mtSSU maturation. In addition, we show that MTG3 can remain bound to the mtSSU via its NTD until the final maturation steps and even during initiation complex assembly. Surprisingly, our data suggest that the latter can commence before the rRNA, and in particular h44, is completely matured. This suggests the existence of several mtSSU biogenesis pathways and, for at least one of them, a coupling of mtSSU

maturation and translation initiation by the action of MTG3, which may act as a quality control factor.

Our biochemical analysis of cells expressing mutant variants of MTG3 provides insights into the molecular function of this factor. In particular, the differential effects of mutations within the NTD and the GTPase domain suggest a potential dual role of MTG3 during mtSSU

**Fig. 5 | Deletion in N-terminus affects MTG3 functionality. a** ΔN-MTG3$^{FLAG}$ lacking 10 conserved aa in the N-terminus can only partially restore mitochondrial translation. Translation of mtDNA-encoded proteins in wild type, *Mtg3$^{-/-}$* and *Mtg3$^{-/-}$* cell lines expressing full length (MTG3$^{FLAG}$) or mutated MTG3 (ΔN-MTG3$^{FLAG}$), respectively, were analyzed using [$^{35}$S]Methionine de novo incorporation and visualized via autoradiography and western blotting. GAPDH was used as a loading control. Similar results were obtained in n ≥ 3 biologically independent experiments. **b** ΔN-MTG3$^{FLAG}$ does not co-purify any MRPs. FLAG-immunoprecipitation was performed with wild type cells, wild type cells expressing MTG3$^{FLAG}$ and *Mtg3$^{-/-}$* cells expressing MTG3$^{FLAG}$ or ΔN-MTG3$^{FLAG}$, respectively. Samples were analyzed using western blotting and antibodies as indicated (total = 3%, eluate = 100%). Similar results were obtained in *n* ≥ 3 biologically independent experiments. **c, d** Monosome level can be restored upon ΔN-MTG3$^{FLAG}$ expression. Mitoplasts (500 μg) were isolated from wild type, *Mtg3$^{-/-}$* and *Mtg3$^{-/-}$* cell lines expressing full length (MTG3$^{FLAG}$) or ΔN-

MTG3$^{FLAG}$, respectively. Mitoribosomal complexes were separated via sucrose density gradient centrifugation and collected fractions (1-16) were analyzed via western blotting with indicated antibodies against MRPs and assembly factors (**c**) or northern blotting with probes against 12S rRNA (*MT-RNR1*) and 16S rRNA (*MT-RNR2*) (**d**). Similar results were obtained in *n* ≥ 3 biologically independent experiments. **e** Abundance of mt-mRNAs bound to monosomes is similar in the *Mtg3$^{-/-}$* + ΔN-MTG3$^{FLAG}$ cell line in comparison to wild type MTG3$^{FLAG}$. Fraction 11 from sucrose gradients from (**c**) were used to isolate RNA and perform NanoString analysis. Level of mt-mRNAs were normalized to 16S-rRNA and RNA abundance bound to monosomes in the *Mtg3$^{-/-}$* + ΔN-MTG3$^{FLAG}$ cell line was calculated relative to *Mtg3$^{-/-}$* + MTG3$^{FLAG}$ cell line (*n* = 3 biologically independent samples shown as mean ± SEM; individual data points are shown as circles). Statistical analysis was performed as two-sample one-tailed Student's *t*-test. Significance was defined as *$p$ ≤ 0.05, **$p$ ≤ 0.01.

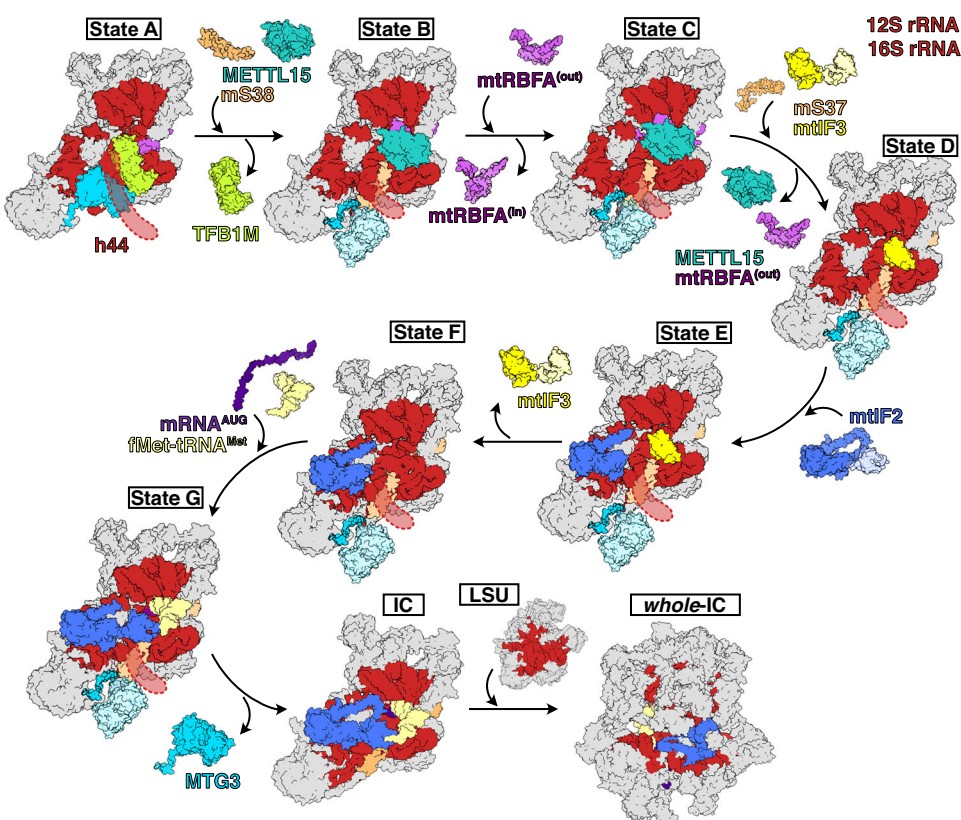

**Fig. 6 | Model of MTG3-mediated maturation coupled to translation initiation of human mtSSU.** Intermediate states of the mtSSU are depicted as surface and factors and rRNA content is colored as in Figs. 2–4. Models for mature mtSSU IC (PDB: 7PO2[19]), mtLSU and mature IC (PDB: 6GAW[53]) have been reported previously.

assembly and maturation: First, MTG3 is bound to the mtSSU with its N-terminus and the globular GTPase domain. GTP hydrolysis might lead to rearrangements in the mtSSU such as h27 maturation, allowing mS38 to bind and displace the globular domain of MTG3. The GTPase activity of MTG3 seems to be essential for continuing mtSSU maturation, as the MTG3-G499P$^{FLAG}$ is unable to form mature mtSSUs. The N-terminus of MTG3 remains stably bound to the mtSSU during late maturation steps and even during the formation of initiation complexes, potentially acting as a quality control mechanism, preventing docking of h44 and therefore premature mtLSU binding. The N-terminal region is not conserved in yeast or bacterial MTG3 homologs, but is highly conserved in other vertebrates, suggesting its function was newly acquired in the course of evolution. Once the mtSSU is matured and primed for translation initiation, the NTD-MTG3 needs to dissociate so that h44 can mature allowing the formation of intersubunit bridges to the mtLSU. Our data do not give insights into what triggers the dissociation of MTG3, we can only speculate that

other, unidentified factors might be involved. In the ΔN-MTG3$^{FLAG}$ cell line, the globular GTPase domain can fulfill its function, so mtSSU maturation can continue and monosomes can be formed. However, without the N-terminus, MTG3 might dissociate after mS38 has displaced the globular domain, and cannot act as a quality control factor. Thus, immature mtSSU particles bypass this critical quality checkpoint leading to premature subunit joining and thus to the formation of translational inactive or less-active mitoribosomes.

The presence of a quality control check point for proper ribosome maturation coupled to translation initiation is reminiscent to the systems described in bacteria, the eukaryotic cytosol, and also in *Trypanosoma brucei* mitochondria. However, the detailed molecular mechanisms differ. In bacteria, IF2 as a GTPase has been discussed as a ribosome assembly factor acting as a chaperone during cold stress[29,30]. The initiator tRNA has been ascribed a direct role during late maturation steps by licensing RNases to perform the final processing steps of the bacterial 16S rRNA[31]. For cytosolic ribosomes, a

translation-like cycle has been reported to monitor the integrity and the ability of the 40S SSU to bind translation factors and the 60S LSU forming 80S-like particles[32]. Final maturation, including the final processing of the rRNA, occurs within these 80S-like complexes. The recruitment of the mature 60S LSU to the immature 40S depends on eIF5B, a translation factor with GTPase activity akin to bacterial IF2. After maturation of the SSU, assembly factors are released and Rli1 dissociates the particles, releasing the mature subunits into the translating pool. These 80S-like ribosomes do not represent translation initiation intermediates as they lack initiator tRNA, mRNA and eIF2. In *Trypanosoma brucei* mitochondria, mtIF2 acts as a quality control factor monitoring the proper folding of the DC while other late assembly factors remain bound to the mtSSU[33]. Together with an mtIF3-like factor, mtIF2 interacts with elements of the DC and blocks the mRNA channel preventing mRNA binding and therefore the formation of a canonical initiation complex. In contrast, we report for human mitochondria an mtSSU complex with bound initiator tRNA, mRNA and mtIF2, but with immature h44 due to the association of MTG3[NTD]. Thus, our data suggest that MTG3 acts as a late quality control factor which remains to be bound until all elements for translation initiation are in place. The release of MTG3 then vacates the binding site of the h44 lasso region and thus licenses the primed initiation complex to enter the translating pool by establishing inter-subunit bridges to the mtLSU.

A common principle in all systems seems to be that domains close to the 3' end, such as h44, are the last regions that are matured[31,33–35]. This also includes RNA modifications such as 12S methylation within h44 and h45 catalyzed by METTL15, NSUN4, and TFB1M[23,24,36]. However, the 12S mt-rRNA requires no further trimming of the 5' or 3' ends after its release from the polycistronic transcript by RNase P and−Z and its assembly with MRPs, which again is distinct from the maturation process described for bacterial and eukaryotic cytosolic SSU. The binding of MTG3 to IC particles might also indicate a function as a splitting factor that dissociates 55S complexes reminiscent to the dual function of GTPBP6, which is required for PTC maturation, but which can also actively dissociate 55S mitoribosomes[11,27]. However, overexpression of MTG3 has no negative impact on mitochondrial translation, which is in contrast to elevated levels of GTPBP6 and thus argues against a function as a ribosome dissociation factor.

In contrast to a previous report, which suggests the dissociation of MTG3 from the mtSSU prior to binding of mS38 and mtRBFA[20], our data indicate that MTG3 remains bound via its NTD to the mtSSU during final maturation steps and even during IC formation. As alternative assembly pathways might occur especially during late assembly steps, these observations are not necessarily in conflict. While it cannot be excluded that ICs with immature rRNA represent an unproductive OFF-pathway, it appears to be a common principle that ribosome maturation is a dynamic process with alternative routes to ensure efficient proper maturation of ribosomes[35,37]. For bacterial ribosomes, for example, it was shown that assembly blocks can join the assembling 50S LSU in a flexible order[37,38]. Under optimal conditions, the most effective pathway is preferred. However, the flexibility ensures that ribosome biogenesis can still continue under non-optimal conditions, such as when a ribosomal protein is depleted, through re-routing of the process. Similarly, alternative pathways that do not require MTG3 might also explain the appearance of pre-SSU states during late assembly and translation initiation, but with folded h44[19,20] and also explain why translation-competent ribosomes can still be formed in *Mtg3*[−/−], although less efficiently (Fig. 1d). In addition, different isolation strategies may lead to the enrichment of different particle populations. While Harper et al.[20] enriched for METTL17-containing complexes via affinity purification of tagged METTL17, and Itoh et al.[19] isolated mitoribosome population from cells deficient in TRMT2B (another known assembly factor of mtSSU), our approach followed the immunoisolation of MTG3-containing complexes. Thus,

depending on the bait and background, different populations from earlier or later maturation steps will be enriched. Taken together, our data suggest an alternative pathway for mitoribosome biogenesis in which MTG3 can remain bound to the mtSSU during the first steps of translation initiation. Whether and under which conditions this pathway is physiologically relevant remains to be determined.

In summary, our work suggests the presence of an alternative mtSSU maturation pathway, where MTG3 has a quality control function during the final maturation steps of the mtSSU by preventing the docking of h44 during the IC formation primed for translation initiation.

## Methods
### Cell culture
HEK293 cell lines (Human Embryonic Kidney 293-Flp-In T-Rex, Thermo Fisher Scientific; R78007) were cultured in DMEM (Dulbecco's Modified Eagle's Medium) supplemented with 10% FCS (Fetal Calf Serum), 2 mM L-glutamine, 1 mM sodium pyruvate and 50 µg/ml uridine under standard cultivation conditions (37 °C under 5% $CO_2$ humidified atmosphere). Cells were regularly tested to be free of Mycoplasma contaminations by GATC Biotech.

To monitor cell growth $7.5 \times 10^4$ cells were seeded into 6-wells plates on day 0 and cell numbers were counted after 1, 2, and 3 days.

HEK293 *Mtg3*[−/−] and *Eral1*[−/−] cell lines were generated by using the Alt-R CRISPR/Cas9 system (Integrated DNA Technologies) according to the manufacturer's instructions. Briefly, cells were co-transfected with Cas9 enzyme and a crRNA-tracrRNA duplex carrying a fluorescent dye to select successfully transfected cells. The crRNAs were designed to target the first exon of *Mtg3* or *Eral1*, respectively. Knockout clones were first tested via immunoblotting and further confirmed via TOPO sequencing. Sequencing of the *Eral1*[−/−] clone revealed premature stop codons in the first exon (Supplementary Fig. 1d), leading to truncated and probably unstable variants of ERAL1.

Stable inducible HEK293 cells lines expressing C-terminal FLAG-tagged MTG3, mutant versions of MTG3 (ΔN-MTG3[FLAG] lacking aa 69-78 and MTG3-G499P[FLAG]), mtIF3 or ERAL1 were generated as described previously[39,40]. Briefly, cells were transfected with pOG44 and pcDNA5/FRT/TO plasmids containing the FLAG construct using Lipofectamine 3000 (Invitrogen) as transfection reagent according to the manufacturer's instructions. Two days after the transfection selection of cells carrying the FLAG construct was started using 100 µg/ml Hygromycin B and 5 µg/ml Blasticidin S.

### Respirometry
For measuring oxygen consumption rates (OCR) and extracellular acidification rates (ECAR), a Seahorse XFe96 Extracellular Flux Analyzer (Agilent) was used according to the manufacturer's instructions. Cells ($5 \times 10^4$ per well) were seeded into a Seahorse XF cell culture plate. For OCR analysis, basal respiration was determined, and subsequently different metabolic conditions were analyzed by adding 3 µM oligomycin, 1.5 µM CCCP, and 0.5 µM antimycin A and rotenone each. For ECAR analysis, acidification of the media was measured under basal conditions and after the addition of 25 mM glucose, 3 µM Oligomycin, and 25 mM 2-deoxy-D-glucose.

### Blue-Native (BN) PAGE and *in gel* activity measurements
For *in gel* activity measurements of complex I and IV, first a BN-PAGE was conducted. Isolated mitochondria were solubilized (1% digitonin, 20 mM Tris-HCl pH 7.4, 0.1 mM EDTA pH 8.0, 50 mM NaCl, 10% glycerol, 1 mM PMSF) at a concentration of 2 µg/µl for 20 min on ice. After centrifugation (20,000 × *g*, 10 min, 4 °C), BN loading buffer (100 mM Bis-Tris pH 7, 500 mM amino caproic acid, 5% Serva Blue G250) was added to the samples, which were subsequently loaded onto a 3–10% polyacrylamide gradient gel and run at 4 °C. Gels were incubated in complex I *in gel* activity solution (2 mM Tris-HCl pH 7.4, 0.1 mg/ml

NADH, 2.5 mg/ml nitro tetrazolium blue) or complex IV solution (0.5 mg/ml diaminobenzidine, 20 µg/ml catalase, 1 mg/ml reduced cytochrome $c$, 75 mg/ml sucrose, 50 mM KP$_i$ pH 7.4), respectively.

## Cell lysates, isolation of mitochondria and mitoplasts preparation

Cell lysis was performed in nonionic lysis buffer (50 mM Tris-HCl pH 7.4, 130 mM NaCl, 2 mM MgCl$_2$, 1% NP-40, 1 mM PMSF, and 1x Protease Inhibitor Cocktail (PI-Mix, Roche)).

For isolation of mitochondria, cells were harvested and resuspended in trehalose buffer (300 mM trehalose, 10 mM HEPES-KOH pH 7.4, 10 mM KCl, 1 mM PMSF, and 0.2% BSA) and homogenized with a Homogenplus Homogenizer (Schuett-Biotech). After removing cell debris (centrifugation at $400 \times g$, 10 min), mitochondria were pelleted at $11,000 \times g$ for 10 min.

To generate mitoplasts, isolated mitochondria were incubated in trehalose buffer with 0.1% digitonin for 30 min on ice following Proteinase K treatment (0.5 µg Proteinase K per 100 µg mitochondria) for 15 min and blocking of the reaction with 2 mM PMSF for 10 min.

## Sucrose gradient ultracentrifugation

Lysed mitoplasts (500 µg in 3% sucrose, 20 mM HEPES-KOH pH 7.4, 100 mM KCl, 20 mM MgCl$_2$, 1x PI-Mix, 0.08 U/µl RiboLock RNase inhibitor (Thermo Fisher) and 1% digitonin) were separated by sucrose gradient ultracentrifugation (5–30% sucrose (w/v) in 20 mM HEPES-KOH pH 7.4, 100 mM KCl, 20 mM MgCl$_2$, 1x PI-Mix) at $79,000 \times g$ for 15 h at 4 °C using a SW41Ti rotor (Beckman Coulter). Fractions (1-16) were collected with a BioComp fractionator, precipitated with 2.5 volumes of ethanol and 1/3 volume of 3 M sodium acetate pH 6.5. Samples were subsequently used for western blotting or RNA was isolated for northern blotting or NanoString analysis.

## Co-immunoprecipitation

Immunoprecipitation of FLAG-tagged proteins was performed as described previously with some modifications[40]. Isolated mitochondria or mitoplasts were lysed in buffer containing 20 mM Tris-HCl pH 7.4 or 20 mM HEPES-KOH pH 7.4, 100 mM NH$_4$Cl or 100 mM KCl, 20 mM MgCl$_2$, 10% Glycerol, 1 mM PMSF, 1x PI-Mix, 1% Triton X-100 and in some cases 0.5 mM GppNHp as indicated. After centrifugation at $16,000 \times g$ at 4 °C for 10 min, the lysate was incubated with anti-FLAG M2 Affinity Gel (Sigma) for 1 h. For the elution of co-purified proteins, FLAG peptides were added. Eluate was either analyzed via western blotting, or used for cryo-EM.

## [$^{35}$S]Methionine de novo synthesis

[$^{35}$S]Methionine labeling of newly synthesized mitochondrial proteins was performed as described previously[40,41]. Cells were incubated in methionine-free media without FCS for 10 min, followed by 10 min incubation in methionine-free media containing 10% FCS and 100 µg/ml emetine to block cytosolic translation. Then 100 µCi/ml [$^{35}$S] Methionine was added and cells were incubated for 1 h. After harvesting, cells were lysed using nonionic lysis buffer (50 mM Tris-HCl pH 7.4, 130 mM NaCl, 2 mM MgCl$_2$, 1% NP-40, 1 mM PMSF and 1x Protease Inhibitor Cocktail (PI-Mix, Roche)) and centrifuged for 2 min at $600 \times g$. Supernatants were collected and protein concentration was determined by Bradford. Protein samples (25 µg) were separated by SDS-PAGE followed by western blotting. Radioactive labeled mitochondrial products were visualized with Typhoon imaging system (GE healthcare).

## RNA isolation and northern blotting

Total RNA or RNA from sucrose gradient fractions was isolated using TRIzol reagent (Life Technologies) or the PureLink RNA Mini Kit (Invitrogen), both according to the manufacturer's instructions. RNA was separated on a denaturing formaldehyde/formamide 1.2% agarose gel and transferred to an Amersham Hybond™-N membrane (GE healthcare) or GeneScreen Plus membrane (PerkinElmer). [$^{32}$P]-radiolabeled probes targeting mitochondrial RNAs were used for visualization with Typhoon imaging system (GE healthcare) (Supplementary Table 4).

## NanoString analysis

RNA was isolated from sucrose gradient fractions and total mitochondrial lysate using TRIzol reagent (Life Technologies) and RNA Clean and Concentrator kit (Zymo Research). RNA was further processed following the manual (NanoString Technologies). In brief, isolated RNA pools were hybridized with TagSet (nCounter Elements XT Reagents, Nanostring Technologies) and specific probes targeting mitochondrial transcripts (Supplementary Table 4)[39,42]. Samples were analyzed with an nCounter MAX system (nanoString Technologies) and data were processed with nSolver software. Raw data of mt-mRNAs were normalized to the abundance of 16S rRNA in the respective fractions.

## Western blotting and immunodetection

Cell lysates, mitochondria samples, or samples recovered from sucrose gradient fractions were separated on 10–18% Tris-Tricine gels and transferred onto nitrocellulose membranes (Cytiva) via semi-dry western blotting. For detection of specific proteins, primary antibodies were incubated overnight at 4 °C following incubation with secondary antibodies coupled to HRP or with fluorescence labeled antibodies for 1 h at room temperature (Supplementary Table 4). Signals were detected using ECL western blotting solution (ThermoFisher Scientific) or LI-COR Odyssey CLx system, and analyzed using Fiji Image J[43].

## Cryo-EM sample preparation and data collection

FLAG-immunoprecipitation was conducted with lysed mitoplasts from a stable HEK293 cell line inducibly expressing MTG3$^{FLAG}$. The eluate was crosslinked with 0.15% glutaraldehyde for 10 min on ice and the reaction was stopped by adding 50 mM lysine pH 7.5 and 50 mM aspartate pH 7.5. The sample was then desalted using Zeba Spin Desalting columns 7 K MWCO (Thermo Scientific) according to the manufacturer's instructions.

For grid preparation, 4 µl of the sample were applied to freshly glow-discharged R 3.5/1 holey carbon grids (Quantifoil) that were precoated with a 2–3 nm carbon layer using a Leica EM ACE600 coater, at 4 °C and 95% humidity in a Vitrobot (FEI). Grids were blotted for 5 s with a blot force of 0 and 60 s before plunge-freezing in liquid ethane.

Cryo-EM data collection was performed with SerialEM using a Titan Krios transmission electron microscope (Thermo Fisher Scientific) operated at 300 keV[44]. Images were acquired in EFTEM mode using a GIF quantum energy filter set to a slit width of 20 eV and a K3 direct electron detector (Gatan) at a nominal magnification of 81,000x corresponding to a calibrated pixel size of 1.05 Å/pixel. Exposures were recorded in counting mode for 3 s with a dose rate of 14.82 e$^-$/px/s resulting in a total dose of 40.33 e$^-$/Å$^2$ that was fractionated into 40 movie frames. Images were acquired in 14 by 1 hole per stage movement.

## Cryo-EM data processing and analysis

Motion correction, CTF-estimation, particle picking, and extraction were performed using Warp[45]. Further processing was carried out using Relion 3.1.0[46] and final non-uniform refinement and local refinement in cryoSPARC[47]. A representative micrograph and the cryo-EM processing workflow are depicted in Supplementary Fig. 2. For initial processing steps, the dataset was split into four batches containing 1,521,397, 1,592,662, 1,569,593 and 1,046,895 4-times binned particles, respectively. From initial 2D classification in Relion, 2,456,047 particles were selected and joined together. From a second round of 2D classification, 663,593 "SSU candidates" and 811,261

"ambiguous SSU" particles were selected. The ambiguous set was re-classified in 2D revealing 242,768 "rescued SSU (1)" particles. The "SSU candidates" were unbinned to a factor of 2 and used for initial 3D auto-refinement and subsequent local 3D classification sorting them into 544,822 good classes ("SSU candidates (1)") and 118,771 "ambiguous SSU (2)". 2D classification of "ambiguous SSU (2)" revealed 53,601 "rescued SSU (2)". All "rescued SSU" sets were joined and unbinned to a factor of 2 followed by 3D auto-refinement and local 3D classification. 78,528 "SSU candidates (2)" were rescued from those and merged with "SSU candidates (1)" ending in 623,350 good SSU particles (10.9% of initially extracted particles) for further processing.

The particles were unbinned and subjected to 3D auto-refinement with a mask around the entire SSU leading to a consensus refinement of 3.0 Å resolution. Two rounds of Bayesian polishing and CTF refinements particles followed by 3D refinement led to a "shiny" reconstruction at 2.8 Å resolution (Supplementary Fig. 2, "**Shiny**"). A head-focus map (Supplementary Fig. 2, "**Head**") at 2.7 Å resolution was generated by subsequent focused 3D refinement with a mask encompassing the head. Signal subtraction with the same mask followed by 3D refinement with a mask around the SSU body generated a body-focused reconstruction without the SSU head (Supplementary Fig. 2, "**body**") at 2.7 Å resolution.

A second round of signal subtraction with a mask around the factor binding site followed by focused 3D classification without image alignment using the same mask revealed subsets containing densities for METTL15, mtRBFA, TFB1M, MTG3, mtIF2, fMet-tRNA$^{Met}$, or empty classes. Re-classification of METTL15 in combination with either mtRBFA$^{(in)}$ or mtRBFA$^{(out)}$ with a mask around METTL15 was performed to increase the signal for METTL15, which was reduced due to its intrinsic flexibility on top of the SSU. All three empty classes were joined together, and all classes were backprojected using the angles from the shiny refinement followed by filtering by local resolution. The final reconstructions of TFB1M and MTG3 (**map A2**), METTL15 and mtRBFA$^{(in)}$ (**map B2**), METTL15 and mtRBFA$^{(out)}$ (**map C2**), mtIF2 and fMet-tRNA$^{Met}$ (**map G2**) are summarized in Supplementary Fig. 2. The joined empty classes and mtIF2-only classes were again subjected to double signal subtraction and local refinements to subtract the SSU head and everything outside a mask encompassing the mtIF3 binding site, revealing three additional classes containing only mtIF3 or mtIF2 or both factors together (Supplementary Fig. 2, **maps D2, E2, F2**). Subsequent non-uniform refinement of these subsets in cryoSPARC[47] led to improved maps of the respective regions (Supplementary Fig. 2, **maps A-G1**). Local refinements using masks encompassing the head, the body, METTL15, mtIF2 (for state F) or mtIF2 and fMet-tRNA$^{Met}$ (for state G) were generated in cryoSPARC (Supplementary Fig. 2, **maps A-G4–6**). For final model building and refinements, composite maps were generated in ChimeraX[48,49] (Supplementary Fig. 2, **maps A-G3**). For states A-C, composite maps were generated using unsharpened local refinement maps to improve visibility of the factors, while the composite maps of state D to G were generated using sharpened local refinement maps.

### Model building and refinement

In order to generate an initial model of the SSU, we rigid-body fitted all residues belonging to the SSU head into the head-focus map (Supplementary Fig. 3, "**Head**") and all residues belonging to body in the global mtSSU reconstruction (Supplementary Fig. 3, "**Shiny**") in ChimeraX, using a published structure from[19] (PDB: 7PO1) as starting model. For state C, which mainly resembles pre-SSU-3a from[19] (PDB: 7PNX), we used this model as a starting point. For state G, which mainly resembles the IC state from[19] (PDB: 7PO2), we used this model as a starting point. The models were manually adjusted and rebuilt in Coot[50] using both unsharpened and postprocessed maps, and subsequently refined in phenix_real_space_refine (PHENIX)[51]. The head and body models were then rigid-body fitted into each of the composite

maps (Supplementary Fig. 3, **maps A-G3**) in ChimeraX followed by manual adjustments in Coot.

In all other states, published structures of factors were used as initial models and could be unambiguously fit into our densities. The models for MTG3, TFB1M, and mtRBFA$^{(in)}$, were taken from[20] (PDB: 8CSP, 8CSR), while mtRBFA$^{(out)}$, METTL15, and mtIF2, mtIF3, and fMet-tRNA$^{Met}$ were taken from[19] (PDBs: 7PNX, 7PO1, 7PO2). All residues within missing or weak density were deleted, and differently folded rRNA content was adjusted to some extent or deleted due to weak densities. Residues in h44 that differ from its mature orientation in states B and C were modeled by rigid-body fit these residues from (PDB: 7PO2) followed by manual adjustments in Coot.

In order to obtain stereochemically sound models, the models for states A-C were interactively re-build and refined using molecular dynamics force fields in ISOLDE[52] within ChimeraX followed by real-space refinements against the composite maps (**map A-C3**) in PHENIX, while the models for states D-G were real-space refined against the composite maps (**map D-G3**) in PHENIX. This resulted in models with excellent stereochemistry. To obtain B-factors that adequately represent the conformational flexibility of head towards body, as well as weaker signal of the factors at the periphery of the whole mtSSU body (Supplementary Tables 1, 2), the models were refined by a final real-space refinement in Phenix against the local B-factor filtered maps (**map A-G2**).

### Reporting summary

Further information on research design is available in the Nature Portfolio Reporting Summary linked to this article.

### Data availability

Material will be available upon request. The electron density reconstructions and structure coordinates were deposited with the Electron Microscopy Database (EMDB), and with the Protein Data Bank (PDB) under accession codes 9G5B and EMD-51083 (State A), 9G5C and EMD-51084 (State B), 9G5D and EMD-51085 (State C), 9HFM and EMD-52117 (State D), 9HFN and EMD-52118 (State E), 9G5E and EMD-51086 (State F), 9HFO and EMD-52119 (State G). The following atomic coordinates were used in this study: 8CSP (human mtSSU assembly intermediate with MCAT, METTL17, TFB1M, MTG3), 8CSR (human mtSSU assembly intermediate with MCAT, METTL17, TFB1M), 7PNX (human mtSSU assembly intermediate with mtRBFA, METTL15), 7PO1 (human mtSSU with mtIF3), 7PO2 (human mtSSU with mtIF2, fMet-tRNA$^{Met}$, mRNA), 3J9M (human mature mitoribosome), 6GAW (human mitoribosome with mtIF2, fMet-tRNA$^{Met}$, mRNA), 6RW5 (human mitochondrial 28S ribosome in complex with mitochondrial IF2 and IF3). Source Data are provided as a Source Data file. Source data are provided with this paper.

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

## Acknowledgements

We thank Christian Dienemann and Ulrich Steuerwald (MPI-NAT cryo-EM facility) for technical assistance during cryo-EM sample preparation and data collection, and Tanja Gall for technical assistance during Seahorse experiments. R.R.-D., H.S.H. and P.R. were supported by the German Research Foundation (DFG, Deutsche Forschungsgemeinschaft) under Germany's Excellence Strategy - EXC 2067/1- 390729940 (to R.R.-D., H.S.H. and P.R.), a DFG Emmy-Noether grant (RI 2715/1-1 to R.R.-D.),

FOR2848 (P10 to H.S.H.), SFB1190 (P23 to H.S.H.), SFB1565 (Project number 469281184, P13 to H.S.H., P14 to P.R., P19 to R.R.-D.) and by the European Union (ERC Starting Grant MitoRNA, grant agreement no. 101116869 to H.S.H. and ERC Advanced Grant MiXpress, ERCAdG no. 101095062 to P.R.). Views and opinions expressed are however those of the author(s) only and do not necessarily reflect those of the European Union or the European Research Council Executive Agency. Neither the European Union nor the granting authority can be held responsible for them. A.F.F. was supported by a Boehringer Ingelheim Fonds Ph.D. fellowship. Funding for open access charge: SFB1565 (Project number 469281184) and Open Access Publication Funds of the Göttingen University.

## Author contributions

Conceptualization, R.R.-D., H.S.H.; Investigation—cell culture, biochemical approaches and sample preparation, M.H., A.K., A.B.; Investigation—cryo-EM sample preparation, data collection, data processing, A.F.F., S.A.; Investigation—building and interpretation structural models, A.F.F., S.A., H.S.H.; Writing—Original Draft, R.R.-D., A.F.F., M.H.; Writing—Review & Editing, R.R.-D., A.F.F., M.H., H.S.H., P.R.; Visualization, A.F.F., M.H.; Supervision, R.R.-D., H.S.H.

## Funding

## Competing interests

The authors declare no competing interests.
