## [Transparent Peer Review file · Nature Communications]

Coupling of ribosome biogenesis and translation initiation in human mitochondria

Corresponding Author: Professor Ricarda Richter-Dennerlein

Version 0:

Reviewer comments:

Reviewer #1

(Remarks to the Author)

In this elegant study, Heinrichs et al. combined genetic disruption and mutagenesis of one of the late assembly factors for mitochondrial ribosomes, GTPase MTG3, with a biochemical and structural analysis. By co-analyzing distinct mt-SSU particles obtained by parallel pulldowns, the authors delivered a deeper resolution to the small subunit assembly process than the one achieved by the previous studies. Their work provides insight into the structural organization and dynamics of the late steps of the small ribosomal subunit (mtSSU) in the mitochondria. It also proposes the functional involvement of particular components of the assembly pipeline in the maturation and quality control checkpoints for mtSSU.

The top novelty of this work is in the demonstration that mitoribosome biogenesis and translation initiation are not separate events, but they can be structurally and timely coupled. Furthermore, it delivers the first evidence that more than one pathway can service the assembly of mitoribosomes, opening a window for follow-up studies on the biological meaning of such coexisting pathways.

The experimental work is well-designed, includes essential controls, and delivers high-quality data supporting the authors' conclusions. It will definitely raise the interest of a vast community studying mitochondrial gene expression and also attract researchers studying ribosome biogenesis in other systems. I highly support the manuscript for publication in Nature Communications, and I have only minor points and questions to address before the ultimate decision:

1. The differences between the current and previous studies focused on mtSSU assembly (Fig.3; Harper et al., 2023) are striking. I agree with the author's conclusion that such discrepancy can indicate the existence of parallel (alternative) pathways for mtSSU assembly possibly featured by distinct assembly kinetics/dynamics. Do the authors think these are the alternative pathways yielding the same output, or if they can serve distinct purposes? For example, can some of these pathways support purely de novo assembly of the mtSSU, and others can be used to fine-tune or store pre-crafted SSUs before their reuse?
2. It was intriguing to see that although ERAL1 is clearly pulled down with MTG3FLAG (Fig.2A), it seems to not end up in the ultimate complex (Fig.2B). Could it be that the interaction of ERAL1 with SSU intermediates occurs but is somewhat transient in the conditions tested by authors? What could be another reason for MTG3-ERAL1 interaction (direct/indirect)? Could authors comment on it in the discussion? Would the reciprocal immunoprecipitation with ERAL1 as a bait give a similar profile to that of the experiment in Fig.2A? Such an experiment would further prove the existence of alternative pathways mediated by a distinct set of machinery (combination of assembly factors).
3. In line and by curiosity: what is the scaffold that stabilizes ERAL1 and TFB1M in fractions 4-5 and 8-9 upon MTG3 KO (Fig.1.I)?
4. Regarding the mechanism for the dissociation of MTG3 from the pre-initiation complex (PIC), it is slightly unclear from the text description of the "coupling model" what exactly triggers the ultimate maturation of h44. Is it methylation, conformational change, recruiting another factor (e.g., fMet-tRNA-Met), or "clicking out" the MTG3? If the last one, what makes MTG3 leave the PIC complex? Could the authors clarify their hypothesis a bit more?
5. The observation that deltaNTD-MTG3 permits translation ATP6/8 while other products are decreased (Fig.5A) is super

interesting. Especially from the perspective of bidirectional ATP synthase activity in a pathological context, but also considering the translational co-regulation of ATP6/8 in yeast (PMID:26823015). Is something known about the tissue expression patterns of MTG3 versus other components of the mtSSU assembly pathway? Or whether particular pathological conditions induce them?

6. Statistical evaluations of presented data quantifications (e.g., Fig.5.E, Fig.1.F) would make authors conclusions even stronger.

Reviewer #2

(Remarks to the Author)

The manuscript by Heinrichs et al. reports the results of a functional and structural investigation poised at unraveling the coupling between the ribosomal biogenesis and translation initiation in the human mitochondrial ribosome (mitoribosome). More specifically, the authors implicate the mitochondrial small ribosomal subunit (mtSSU) maturation factor, MTG3 in this process and attempt to demonstrate that the mtSSU biogenesis and translation initiation can occur simultaneously, thus providing new insight on this functional coupling.

The manuscript is well written, the figures are very clear and even aesthetic. The references are sufficient and the message of the manuscript is streamlined.

The reviewer has few minor and major concerns.

Minor concerns:

In Results. The authors write “The loss of MTG3 significantly affects cell growth and is accompanied by rapid acidification of the media even with high-glucose, indicating a mitochondrial dysfunction(Fig. 1c, data not shown)”. Although I trust the authors’ claims, it would be useful to show this data, if possible.

In Results. The authors write cite the used method for monitoring mitochondrial translation. The authors cite their 2018 reference in NAR for the methodology, along with the Anne Chomyn 1996 Methods in Enzymology paper. In their NAR reference they cite two other papers, one of which is the same methods in enzymology paper, but to follow the entire protocol the reader will have to chase down the long string of auto-references and its simply very time consuming, especially for reviewers who are not planning to replicate the experimental procedure... Could the authors please in few lines at the Methods section explain the general principle of how this was done? How were the mtDNA encoded proteins purified for their analysis after incorporation of Methionine S35? by immunoprecipitation?

In Results, page 8, line 265-266: The authors write “our data suggest that IC intermediates co-purify with the assembly factor MTG3 and vice versa”.

This contrasts with the work by Itho et al., 2022 where they show that the last step of SSU assembly passes the baton to the first step of pre-initiation through IF3 that then remains bound to RBFA in the out conformation before the latter is replaced by mS37, could the authors comment on this? This is very tricky especially when h44 is not correctly folded.

In Results, page 8, line 271-272: The authors write “State D resembles the previously described structure of the PIC with mtIF3 bound, which we could fit into the density without readjustments”.

I find this statement a bit tricky and to some extent misleading, as the previously described work the authors refer to (Itho et al., 2022) shows a state that is indeed partially similar to this described state D, however h44 seems to be fully folded and as far as one could tell from the cited article, no traces of MTG3 NTD. Could the authors comment?

In Results, page 8, line 275-276: The authors write the following referring to the State G “In this state, we observe clear density for the codon-anticodon interaction, indicating that it represents an IC primed for translation initiation”.

In all these (pre)initiation states h44 is misfolded, or at least immature observed along with that of MTG3. This is a concern as at this stage MTG3 should have left after the maturation of h44. This compromises any strong conclusions from the observed structures as it is possible that none of them is physiologically relevant

In Results, page 8, line 281-283: The authors write “This suggests that h44 maturation is not required for IC formation and that translation initiation factors can be recruited before mtSSU maturation is finalized. However, the association of MTG3 during IC formation prevents docking of h44 and thus mtLSU recruitment to the mtSSU”.

Very strong statement, as there are no clear indications that translation initiation can be completed without the full maturation of h44. at least up to a certain stage, perhaps the authors want to rephrase.

Major concerns:

In Results, page 3, line 98-100: The authors write "Interestingly, we observe a differential reduction in multiple MRPs of the mtSSU. The mt-rRNA-dependent MRPs uS14m and uS15m and the late binding protein mS37 are drastically decreased to 20-30%, whereas other".

Logically speaking, the variable expression level of nuclear encoded MRPs doesn't indicate the role of MTG3 in late maturation, as no direct causality between the level of expression of this maturation factor and the expression level of other MRPs can be characterized in the current or prior data. It would be extremely interesting to unveil and understand a feedback signal from the assembly defects of the SSU that can retro regulate the expression level of several MRPs... The authors should comment further about such causality or rephrase. It is the reviewer's opinion that the question of rather the stability of these MRPs (that assemble at a late stage) in the context of imperfect maturation of the mtSSU, i.e., when they are free off the maturing mtSSU. The full maturation of the mtSSU and the recruitment of these late stage MRPs would result in their protection after their binding, probably. But in no way, unless proven otherwise, the authors should phrase a putative link between the expression of MTG3 and the EXPRESSION (suggesting transcription and translation of the concerned MRPs) of several MRPs.

In Results, page 3, line 105-106: The authors write "we observe a significant reduction of 12S mt rRNA to 40%, whereas the 16S mt-rRNA remains stable".

Similar to the previous concern regarding the MRPs, the same holds true for rRNA levels. What is the causality between the expression of MTG3 and the EXPRESSION of the 12S rRNAs? Perhaps the authors mean to speak of the "stability of the rRNA" in which case they should spell it out clearly. In Results, page 10, line 351-355: The authors write "Our structural and biochemical data suggest that mtSSU assembly and translation initiation are two processes that do not stringently occur sequentially after each other, and allow us to deduce an alternative model of mtSSU biogenesis".

The authors did a great job in describing their complexes and structures, but the idea that translation initiation complexes can proceed without being fully mature still seems a bit odd. If the binding sites of the different initiation factors are well folded and in the correct conformations then one would expect the different initiation factors to bind on these sites. However, if other sites of the mtSSU are not fully mature such as the essential h44 that forms several bridges with the LSU, then translation can't proceed. In this work, it is unknown whether these trapped complexes can finish their maturation before translation can proceed or they are OFF PATHWAY, which is frequently observed in other systems such as the cytosolic SSU maturation intermediates as thoroughly inspected by the Beckmann and Hurt labs, where they show the existence of numerous off pathway assembly intermediates that accumulate and probably degrade without yielding mature complexes. The reviewer's thinking is the following; if these complexes were able to proceed, in the absence of any blocking agents, then they would be extremely short living and won't be capturable by cryo-EM, unless more advanced methods such as time-resolved were applied. Nevertheless, this doesn't mean that the presented structures are meaningless! The presented structures, by extrapolation, provide a view on some of the late-stage intermediate steps.

Reviewer #3

(Remarks to the Author)

This manuscript uses mutational, biochemical, and single-particle cryo-EM analysis to analyze assembly intermediates of the human small mitoribosomal subunit (mtSSU). The authors generate a MTG3 GTPase ablated cell line to show that reduction of MTG3 function causes faulty mitochondrial translation, which is restored by expression of FLAG-tagged wild-type MTG3. They also show mitochondrial ribosomal proteins (MRPs) uS14m, uS15m, and mS37 to be reduced by 70-80% in the ablated MTG3 condition. They use the FLAG-tagged MTG3 to isolate MTG3-containing mtSSU complexes by co-immunoprecipitation, which also show the presence of translation initiation factors. They additionally perform mutational analysis of MTG3 and its effects on mtSSU generation. Based on these results, they suggest that mtSSU biogenesis and translation initiation occur simultaneously in human mitochondria, and are functionally coupled.

The manuscript is written well and the results are interesting but I do have the following concerns:

1. It is not immediately clear to me (and therefore might not be immediately clear to a general audience) how the FLAG-tag purification of the mtSSU complexes is done exactly. The phrasing in the Figure legends is 'FLAG-immunoprecipitation was performed with lysed mitochondria from wild type cells and cells inducibly expressing MTG3FLAG,'. Is a purified FLAG-tagged protein used to isolate the mtSSU complexes from wild-type cells, which would require some incubation with the protein and the mitoplast extract? Is an internally expressed FLAG-tagged MTG3 used to purify MTG3-FLAG-mtSSU complexes that were formed in vivo? What is the exact source of the complexes for which the cryo-EM data has been generated (wild-type or inducibly expressed FLAG-MTG3 cells)? I may have missed this information but it was clearly hard for me to find.
2. The cryo-EM samples have been generated by crosslinking with glutaraldehyde, which could result in artificially enhanced occupancy of proteins on the mtSSU. It is not clear why this was done, this choice should be explained. The title of the paper and its conclusions that assembly and initiation are coupled is too definitive considering this aspect of the experimental design.

3. The PDB models have been generated for only four density maps. I am not convinced that depositing PDB models for the other states being described can be avoided. The authors can deposit the models that they have generated by rigid body fitting and mention that fact in the deposition. I don't see the point of making readers replicate such fitting if they wish to compare multiple assembly states being described.

4. The resolution of the maps is not very high, the review would be more informed if the reviewers could see the quality of the maps and the models fit into them for themselves. For example, the structures where only the MTG3 N-terminal domain (NTD) is bound, and the C-terminal domain has dissociated, are not at very high resolution. In spite of MTG3-FLAG purification being used, the putative NTD density could belong to some other protein. The sidechain densities for this region being unambiguous would have countered this possibility, but it is not clear if they are.

5. The figures depicting structures seem excessively complicated. A jumble of colors makes it harder for readers to follow exactly what is being indicated. It is better to simplify them. For example, showing the ribosomal RNA can be dispensed with in multiple figure panels. Showing the full ribosome with multiple magnified views extracted within the same panel could be avoided. There might be other ways to avoid the visual clutter.

6. Line 383, replace 'letter' with 'latter'.

Version 1:

Reviewer comments:

Reviewer #1

(Remarks to the Author)

The authors have addressed all my points and suggestions comprehensively. Additional controls, quantifications, and explanations cleared up my previous questions. I also appreciate new visualizations of the structures that improved the clarity of figures. Although the biological meaning of the model proposed by authors remains to be determined, coupling the mitoribosome biogenesis to the early steps of translation initiation is a new concept with interesting implications for mitochondrial function. It also expands on the role of MTG3. With that, I support the work by Heinrichs et al. for publication in Nature Communications.

Reviewer #2

(Remarks to the Author)

The authors have addressed appropriately all the critical points raised by the reviewer.

Reviewer #3

(Remarks to the Author)

The authors have made the changes required to address my previous comments. They have also modeled the states not previously modeled and deposited those models as well. I do have a few minor comments after examining the maps and models that they have kindly provided:

map A: 12SRNA residues helical hairpin residues 891-908 are not modeled but there seems to be density present for them, part of which shows up as unmodeled density even at high thresholds.

map B: There is unmodeled density near 12S rRNA residues helical hairpin residues 890-910. 12S rRNA h44 residues 1502-1548 not modeled, there is some additional clear double helical density that can be modeled.

map C: 12S RNA h44 residues 1502-1550 not modeled but there is some additional clear double helical density that can be modeled.

map D-F: Some extra poorly-resolved density on h44 visible that likely cannot be modeled accurately.

map G: Some extra poorly-resolved density on h44 and the mtlF2 domain interacting with tRNA. It may be worth providing a C-alpha model for that mtlF2 domain to show its overall position.

I am attaching a figures showing the unmodeled density as red for states A-G.

[Editorial Note: This figure attachment is displayed on the final page of this file]

The simplified structure figures are indeed clearer in my opinion. Some minor thoughts on those:

Fig. 2c: Location of boxes in the full small subunit images does not always seem to be the same as the enlarged panels d-i. The box labeled d has an extra vertical line.

Fig. 3c: Is it corresponding to the box in the full small subunit image in Fig. 2b?

Version 2:

Reviewer comments:

Reviewer #3

(Remarks to the Author)

The changes made by the authors in response to my previous comments are satisfactory. They have clarified where they could make improvements and justified the reasons for where they could not. Among the text changes, I am not quite sure what this last phrase means in lines 658-659: "where visibility of the factors is nevertheless given."

Point-by-point response to the reviewers' comments

Reviewer #1 (Remarks to the Author):

In this elegant study, Heinrichs et al. combined genetic disruption and mutagenesis of one of the late assembly factors for mitochondrial ribosomes, GTPase MTG3, with a biochemical and structural analysis. By co-analyzing distinct mt-SSU particles obtained by parallel pulldowns, the authors delivered a deeper resolution to the small subunit assembly process than the one achieved by the previous studies. Their work provides insight into the structural organization and dynamics of the late steps of the small ribosomal subunit (mtSSU) in the mitochondria. It also proposes the functional involvement of particular components of the assembly pipeline in the maturation and quality control checkpoints for mtSSU.

The top novelty of this work is in the demonstration that mitoribosome biogenesis and translation initiation are not separate events, but they can be structurally and timely coupled. Furthermore, it delivers the first evidence that more than one pathway can service the assembly of mitoribosomes, opening a window for follow-up studies on the biological meaning of such coexisting pathways.

The experimental work is well-designed, includes essential controls, and delivers high-quality data supporting the authors' conclusions. It will definitely raise the interest of a vast community studying mitochondrial gene expression and also attract researchers studying ribosome biogenesis in other systems. I highly support the manuscript for publication in Nature Communications, and I have only minor points and questions to address before the ultimate decision:

We highly appreciate the positive feedback by the reviewer.

R1-1

1. The differences between the current and previous studies focused on mtSSU assembly (Fig.3; Harper et al., 2023) are striking. I agree with the author's conclusion that such discrepancy can indicate the existence of parallel (alternative) pathways for mtSSU assembly possibly featured by distinct assembly kinetics/dynamics. Do the authors think these are the alternative pathways yielding the same output, or if they can serve distinct purposes? For example, can some of these pathways support purely de novo assembly of the mtSSU, and others can be used to fine-tune or store pre-crafted SSUs before their reuse?

Although this is an interesting question, we can only speculate. Under optimal circumstances there might be one energetically favourable route, however, under perturbed conditions alternative pathways, which are less efficient, are pursued. Similar observations were made e.g. for the bacterial LSU, where depletion of uL18 revealed the accumulation of assembly intermediates which are competent for maturation. Here, the authors revealed that the process of subunit assembly is highly dynamic and can be re-routed under defined conditions. We extended the discussion in this context.

R1-2

2. It was intriguing to see that although ERAL1 is clearly pulled down with MTG3FLAG (Fig.2A), it seems to not end up in the ultimate complex (Fig.2B). Could it be that the interaction of ERAL1 with SSU intermediates occurs but is somewhat transient in the conditions tested by authors? What could be another reason for MTG3-ERAL1 interaction (direct/indirect)? Could authors comment on it in the discussion? Would the reciprocal immunoprecipitation with ERAL1 as a bait give a similar profile to that of the experiment in Fig.2A? Such an experiment

would further prove the existence of alternative pathways mediated by a distinct set of machinery (combination of assembly factors).

Indeed, performing the reciprocal experiment using ERAL1^{FLAG} as a bait results in the co-isolation of MTG3. We included the additional experiment in the manuscript (New Supplemental Fig. 1e).

Furthermore, we included a co-immunoprecipitation of MTG3-G499P^{FLAG} (Supplementary Fig. 6e). This GTPase mutant variant is still able to interact with early-binding MRPs such as mS27, mS40 and uS17m as well as with ERAL1. Thus, association of ERAL1 to the assembling mtSSU occurs prior to GTP-hydrolysis by MTG3. However, abolishing GTP hydrolysis of MTG3 prevents the association of late-binding proteins such as mS37 or the assembly factor mtRBFA, highlighting that the GTPase activity is required for mtSSU biogenesis and that MTG3 is part of several assembly states.

The reason why ERAL1 was not part of our solved states might be due to the flexibility of GTPases. This seems to be a common observation as also other GTPases, although purified together with ribosome particles, are difficult to solve via cryo-EM (e.g. GTPBP7) (Lenarčič et al., 2021). Moreover, it is important to consider that our lack of observation of ERAL1-bound mtSSUs does not necessarily mean that these particles do not exist in our sample, but they appear to represent a minor population with not enough particles to allow for classification and reconstruction. While Harper et al. (2023) show ERAL1 and MTG3 in one complex, these states were co-purified via METTL17. Under these conditions the yield of earlier intermediates with ERAL1 might be higher and the GTPase in a more stabilised state. Vice versa, other complexes which we solved via MTG3 purification could not be solved by Harper et al., indicating that the purification strategy and bait can have a strong influence on which intermediates can be enriched and solved by cryo-EM. We have included this in the discussion.

R1-3

3. In line and by curiosity: what is the scaffold that stabilizes ERAL1 and TFB1M in fractions 4-5 and 8-9 upon MTG3 KO (Fig.1.I)?

This is an interesting observation. We assume that the reviewer refers to fractions 6/7, where ERAL1 and TFB1M are still detectable in the MTG3 knockout. MTG3 ablation does not lead to a complete loss of mtSSU. Residual levels are still detectable in fractions 6/7 and 11 (see mS40 and uS17m). These residual levels of mtSSU might present assembly intermediates stalled at the state where ERAL1 and TFB1M are bound. Thus, without MTG3 these assembly intermediates cannot progress further and accumulate, as discussed in the text.

R1-4

4. Regarding the mechanism for the dissociation of MTG3 from the pre-initiation complex (PIC), it is slightly unclear from the text description of the “coupling model” what exactly triggers the ultimate maturation of h44. Is it methylation, conformational change, recruiting another factor (e.g., fMet-tRNA-Met), or “clicking out” the MTG3? If the last one, what makes MTG3 leave the PIC complex? Could the authors clarify their hypothesis a bit more?

The ultimate maturation of h44 requires the dissociation of MTG3 as the N-terminus of MTG3 prevents the docking of h44 to the foot of the mtSSU. At this point we do not know what triggers the dissociation of MTG3. We can only speculate that another, yet unknown factor might be involved, which we now included in the discussion. However, we refrain from putting too much emphasis on this as our data do not give insights into this process.

R1-5

5. The observation that deltaNTD-MTG3 permits translation ATP6/8 while other products are decreased (Fig.5A) is super interesting. Especially from the perspective of bidirectional ATP synthase activity in a pathological context, but also considering the translational co-regulation of ATP6/8 in yeast (PMID:26823015). Is something known about the tissue expression patterns of MTG3 versus other components of the mtSSU assembly pathway? Or whether particular pathological conditions induce them?

We agree with the reviewer that the restored translation of ATP6/8 in the dN-MTG3 is particularly interesting, especially considering the reversed ATP synthase activity to preserve the mitochondrial membrane potential under pathological conditions. To our knowledge, MTG3 is ubiquitously expressed, however, we cannot exclude that MTG3 expression is altered under certain pathological conditions, which needs to be addressed in future studies.

Although, we agree with the reviewer that it is important to consider other systems such as yeast, the translation systems in human and yeast mitochondria differ substantially. Yeast employs specific translation activators and repressors to regulate mitochondrial translation, while human mitochondria evolved different mechanisms to adapt the function of their translation machinery lacking factors homologous to yeast translation activators.

*Additionally, the N-terminal domain of MTG3 is not conserved in yeast (see **Supplementary Fig. 6a**), which also indicates that mtSSU assembly and translation regulation differ mechanistically between yeast and human mitochondria. Due to these differences, we refrain from speculating about translational co-regulation of ATP6/8 based on data obtained in the yeast system.*

R1-6

6. Statistical evaluations of presented data quantifications (e.g., Fig.5.E, Fig.1.F) would make authors conclusions even stronger.

*We included statistical evaluations for **Figures 1c, 1f, 1h, 5e, and Supplementary Fig. 6b**.*

Reviewer #2 (Remarks to the Author):

The manuscript by Heinrichs et al. reports the results of a functional and structural investigation poised at unraveling the coupling between the ribosomal biogenesis and translation initiation in the human mitochondrial ribosome (mitoribosome). More specifically, the authors implicate the mitochondrial small ribosomal subunit (mtSSU) maturation factor, MTG3 in this process and attempt to demonstrate that the mtSSU biogenesis and translation initiation can occur simultaneously, thus providing new insight on this functional coupling.

The manuscript is well written, the figures are very clear and even aesthetic. The references are sufficient and the message of the manuscript is streamlined.

We thank the reviewer for this positive feedback.

The reviewer has few minor and major concerns.

Minor concerns:

R2-1

In Results. The authors write “The loss of MTG3 significantly affects cell growth and is accompanied by rapid acidification of the media even with high-glucose, indicating a mitochondrial dysfunction(Fig. 1c, data not shown).”. Although I trust the authors’ claims, it would be useful to show this data, if possible.

*We included measurements for oxygen consumption rate (OCR) and extracellular acidification rate (ECAR) as well as in gel activity analysis for complex I and IV which clearly demonstrate mitochondrial dysfunction in Mtg3^{-/-}. We included the data as new **Supplementary Fig. 1b-c**.*

R2-2

In Results. The authors write cite the used method for monitoring mitochondrial translation. The authors cite their 2018 reference in NAR for the methodology, along with the Anne Chomyn 1996 Methods in Enzymology paper. In their NAR reference they cite two other papers, one of which is the same methods in enzymology paper, but to follow the entire protocol the reader will have to chase down the long string of auto-references and its simply very time consuming, especially for reviewers who are not planning to replicate the experimental procedure... Could the authors please in few lines at the Methods section explain the general principle of how this was done? How were the mtDNA encoded proteins purified for their analysis after incorporation of Methionine S35? by immunoprecipitation?

We included a detailed description in the methods section:

“Cells were incubated in methionine-free media without FCS for 10 min, followed by 10 min incubation in methionine-free media containing 10% FCS and 100 µg/ml emetine to block cytosolic translation. Then 100 µCi/ml [³⁵S]Methionine was added and cells were incubated for 1 h. After harvesting, cells were lysed using nonionic lysis buffer (50 mM Tris-HCl pH 7.4, 130 mM NaCl, 2 mM MgCl₂, 1 % NP-40, 1 mM PMSF and 1x Protease Inhibitor Cocktail (PI-Mix, Roche)) and centrifuged for 2 min at 600 xg. Supernatants were collected and protein concentration was determined by Bradford. Protein samples (25 µg) were separated by SDS-PAGE followed by western blotting. Radioactive labelled mitochondrial products were visualized with Typhoon imaging system (GE healthcare).”

R2-3

In Results, page 8, line 265-266: The authors write “our data suggest that IC intermediates co-purify with the assembly factor MTG3 and vice versa”.

This contrasts with the work by Itho et al., 2022 where they show that the last step of SSU assembly passes the baton to the first step of pre-initiation through IF3 that then remains bound to RBFA in the out conformation before the latter is replaced by mS37, could the authors comment on this? This is very tricky especially when h44 is not correctly folded.

Based on our structural data, we do not believe our observations and those reported by Itho et al are in conflict. In the structures reported here, the part of h44 crucial for the decoding center (A- and P-site) is already matured, whereas the lasso region/ bottom part of h44 cannot dock to the foot of the mtSSU in the presence of the NTD of MTG3, preventing the formation of intersubunit bridges. Also, the fact that we observe codon-anticodon base-pairs in state G argues for a matured decoding center even if the bottom part of h44 is not placed in its final position. Thus, the association of mtIF3 and mS37 can occur. It is also not possible to exclude that there is a complex of mtSSU+mtRBFA+mtIF3+MTG3 in our co-isolation, which we may not be able to detect if it represents a minor fraction.

Additionally, as we discussed in the text, there might be multiple alternative assembly routes, which ensure efficient ribosome biogenesis and subunit maturation. The enrichment for certain

states highly depends on the genetic background, perturbation conditions and purification procedure

R2-4

In Results, page 8, line 271-272: The authors write “State D resembles the previously described structure of the PIC with mtIF3 bound, which we could fit into the density without readjustments”.

I find this statement a bit tricky and to some extent misleading, as the previously described work the authors refer to (Itho et al., 2022) shows a state that is indeed partially similar to this described state D, however h44 seems to be fully folded and as far as one could tell from the cited article, no traces of MTG3 NTD. Could the authors comment?

We thank the reviewer for pointing this out. Indeed, this comment is misleading and we adjusted the text accordingly. In addition, we now also provide and deposit PDB models for state D to G in the revised manuscript, as also requested by reviewer #3.

R2-5

In Results, page 8, line 275-276: The authors write the following referring to the State G “In this state, we observe clear density for the codon-anticodon interaction, indicating that it represents an IC primed for translation initiation”.

In all these (pre)initiation states h44 is misfolded, or at least immature observed along with that of MTG3. This is a concern as at this stage MTG3 should have left after the maturation of h44. This compromises any strong conclusions from the observed structures as it is possible that none of them is physiologically relevant

*We agree with the reviewer that the physiological relevance of the states described in this manuscript will need to be further investigated in the future. However, we do believe that the observations in this manuscript warrant drawing conclusions from them for several reasons. First, these states were observed in samples of endogenous complexes from HEK293 cells inducibly expressing MTG3^{FLAG}. The affinity labelling of specific factors is a common strategy in the field which has also been used in other studies investigating the mechanisms of mitochondrial ribosome assembly (e.g. Harper et al. 2023, where METTL17 was tagged). Second, the expression of MTG3^{FLAG} has no negative effect on mitochondrial translation (**see e.g. Fig. 5a**) indicating that we do not shift the equilibrium and do not stall assembly intermediates or accumulate OFF pathway complexes. This is quite often a concern when working with genetically perturbed cell lines deficient in e.g. essential assembly factors. (see also our comment to R2-9). Third, as described in our response to comment R2-3, there is no obvious structural or mechanistic reason why preinitiation complexes should not be able to form while parts of h44 are misfolded, as the regions forming the decoding center appear to be already matured.*

Taken together, while we agree with the reviewer that the observation of these states is somewhat unexpected, we have no reason to believe that this compromises their relevance. Instead, their observation enables us to propose a refined model for the maturation of the mtSSU. In the revised manuscript, we have included a section in the discussion emphasizing the uncertainty regarding the physiological relevance of the observed states.

R2-6

In Results, page 8, line 281-283: The authors write “This suggests that h44 maturation is not required for IC formation and that translation initiation factors can be recruited before mtSSU

maturation is finalized. However, the association of MTG3 during IC formation prevents docking of h44 and thus mtLSU recruitment to the mtSSU”.

Very strong statement, as there are no clear indications that translation initiation can be completed without the full maturation of h44. at least up to a certain stage, perhaps the authors want to rephrase.

*We agree with the reviewer that translation initiation cannot be **completed** without the full maturation of h44, but IC formation can be initiated and progress to a certain stage.*

We have rephrased the text to make this point more clear.

Major concerns:

R2-7

In Results, page 3, line 98-100: The authors write “Interestingly, we observe a differential reduction in multiple MRPs of the mtSSU. The mt-rRNA-dependent MRPs uS14m and uS15m and the late binding protein mS37 are drastically decreased to 20-30%, whereas other”.

Logically speaking, the variable expression level of nuclear encoded MRPs doesn't indicate the role of MTG3 in late maturation, as no direct causality between the level of expression of this maturation factor and the expression level of other MRPs can be characterized in the current or prior data. It would be extremely interesting to unveil and understand a feedback signal from the assembly defects of the SSU that can retro regulate the expression level of several MRPs... The authors should comment further about such causality or rephrase.

It is the reviewer's opinion that the question of rather the stability of these MRPs (that assemble at a late stage) in the context of imperfect maturation of the mtSSU, i.e., when they are free off the maturing mtSSU. The full maturation of the mtSSU and the recruitment of these late stage MRPs would result in their protection after their binding, probably. But in no way, unless proven otherwise, the authors should phrase a putative link between the expression of MTG3 and the EXPRESSION (suggesting transcription and translation of the concerned MRPs) of several MRPs.

We apologize for causing confusions. The reviewer is absolutely right. We did not intend to suggest that MTG3 has any influence on the expression per se on MRPs. We rather suggest that certain MRPs are less stable if they are not incorporated in complexes due to assembly defects. We substituted the term “expression levels” with “protein steady state levels”.

R2-8

In Results, page 3, line 105-106: The authors write “we observe a significant reduction of 12S mt rRNA to 40%, whereas the 16S mt-rRNA remains stable”.

Similar to the previous concern regarding the MRPs, the same holds true for rRNA levels. What is the causality between the expression of MTG3 and the EXPRESSION of the 12S rRNAs? Perhaps the authors mean to speak of the "stability of the rRNA" in which case they should spell it out clearly.

We agree and rephrased the text accordingly.

R2-9

In Results, page 10, line 351-355: The authors write “Our structural and biochemical data suggest that mtSSU assembly and translation initiation are two processes that do not stringently occur sequentially after each other, and allow us to deduce an alternative model of mtSSU biogenesis”.

The authors did a great job in describing their complexes and structures, but the idea that translation initiation complexes can proceed without being fully mature still seems a bit odd. If the binding sites of the different initiation factors are well folded and in the correct conformations then one would expect the different initiation factors to bind on these sites. However, if other sites of the mtSSU are not fully mature such as the essential h44 that forms several bridges with the LSU, then translation can't proceed. In this work, it is unknown whether these trapped complexes can finish their maturation before translation can proceed or they are OFF PATHWAY, which is frequently observed in other systems such as the cytosolic SSU maturation intermediates as thoroughly inspected by the Beckmann and Hurt labs, where they show the existence of numerous off pathway assembly intermediates that accumulate and probably degrade without yielding mature complexes.

The reviewer's thinking is the following; if these complexes were able to proceed, in the absence of any blocking agents, then they would be extremely short living and won't be capturable by cryo-EM, unless more advanced methods such as time-resolved were applied. Nevertheless, this doesn't mean that the presented structures are meaningless! The presented structures, by extrapolation, provide a view on some of the late-stage intermediate steps.

*Structures and IP data suggest that initiation factors indeed bind to the mtSSU intermediates (State D-F) while h44 is still immature, meaning that the translation initiation phase can be initiated. However, we do not wish to argue that translation initiation can continue with immature h44. Of course, MTG3 needs to dissociate before the mtLSU can bind and thus translation initiation can continue. We cannot completely rule out that our captured complexes might represent OFF pathways, but as all parts except the lasso region of h44 comprise the mature conformation of an IC complex in state G, including codon-anticodon base pairing in the P-site, we assume that they could be translation-competent after MTG3 dissociation. Additionally, overexpression of MTG3 does not affect mitochondrial translation, indicating that it does not shift the equilibrium to potential OFF pathways in the corresponding cell line. We also would like to emphasize that we do not perturb mitoribosome biogenesis by depleting an essential assembly factor, which indeed often results in OFF pathways. Here, we simply enrich for MTG3-containing complexes via FLAG-co-immunoprecipitation. The FLAG tagged variant of MTG3 can restore mitochondrial translation in *Mtg3*^{-/-}, indicating that MTG3^{FLAG} is functional. Overall, while we cannot exclude the possibility that the states we observe represent an OFF pathway, we do not think this is the most likely explanation of the data. However, we have extended the discussion to account for the possibility of an OFF pathway.*

Reviewer #3 (Remarks to the Author):

This manuscript uses mutational, biochemical, and single-particle cryo-EM analysis to analyze assembly intermediates of the human small mitoribosomal subunit (mtSSU). The authors generate a MTG3 GTPase ablated cell line to show that reduction of MTG3 function causes faulty mitochondrial translation, which is restored by expression of FLAG-tagged wild-type MTG3. They also show mitochondrial ribosomal proteins (MRPs) uS14m, uS15m, and mS37 to be reduced by 70-80% in the ablated MTG3 condition. They use the FLAG-tagged MTG3 to isolate MTG3-containing mtSSU complexes by co-immunoprecipitation, which also show the presence of translation initiation factors. They additionally perform mutational analysis of MTG3 and its effects on mtSSU generation. Based on these results, they suggest that mtSSU

biogenesis and translation initiation occur simultaneously in human mitochondria, and are functionally coupled.

The manuscript is written well and the results are interesting but I do have the following concerns:

We thank the reviewer for the positive feedback.

R3-1

1. It is not immediately clear to me (and therefore might not be immediately clear to a general audience) how the FLAG-tag purification of the mtSSU complexes is done exactly. The phrasing in the Figure legends is 'FLAG-immunoprecipitation was performed with lysed mitochondria from wild type cells and cells inducibly expressing MTG3FLAG,'. Is a purified FLAG-tagged protein used to isolate the mtSSU complexes from wild-type cells, which would require some incubation with the protein and the mitoplast extract? Is an internally expressed FLAG-tagged MTG3 used to purify MTG3-FLAG-mtSSU complexes that were formed in vivo? What is the exact source of the complexes for which the cryo-EM data has been generated (wild-type or inducibly expressed FLAG-MTG3 cells)? I may have missed this information but it was clearly hard for me to find.

We apologize for causing confusion. We have used a stable cell line inducibly expressing a FLAG tagged variant of MTG3. The construct for expressing MTG3 with C-terminal FLAG tag was inserted into the Flp-In cassette in the HEK293 cell line. The expression was induced for 24h prior to isolation of mitochondria. Subsequently, mitochondria were lysed and subjected to FLAG-immunoprecipitation to purify endogenous MTG3-containing complexes. We changed the text and figure legend accordingly.

R3-2

2. The cryo-EM samples have been generated by crosslinking with glutaraldehyde, which could result in artificially enhanced occupancy of proteins on the mtSSU. It is not clear why this was done, this choice should be explained. The title of the paper and its conclusions that assembly and initiation are coupled is too definitive considering this aspect of the experimental design.

We agree with the reviewer that the use of crosslinker should be explained in more detail. We initially performed the purification and subsequent structural analysis without crosslinking. Processing of this data revealed the same states as described for the crosslinked dataset, but with lower number of particles in particular for states A-C and therefore limited resolution for many factors. Importantly however, we also observed a partially immature h44 in all states in the uncrosslinked dataset. To improve the recovery of useful particles, we repeated the structural analysis with crosslinking just prior to grid preparation, a strategy that is not uncommon in single-particle cryo-EM to stabilize potentially transient samples. This led to and improved overall occupancy of factors and resolution of the maps, and we thus chose to proceed with the crosslinked dataset for publication.

R3-3

3. The PDB models have been generated for only four density maps. I am not convinced that depositing PDB models for the other states being described can be avoided. The authors can deposit the models that they have generated by rigid body fitting and mention that fact in the deposition. I don't see the point of making readers replicate such fitting if they wish to compare multiple assembly states being described.

*We agree with the reviewer and have now generated models for each state, which we deposited to the PDB. We also attached them for the reviewers (see link).
<https://owncloud.gwdg.de/index.php/s/32gMK7zkPUrA11k> (password mtg3_2024)*

R3-4

4. The resolution of the maps is not very high, the review would be more informed if the reviewers could see the quality of the maps and the models fit into them for themselves. For example, the structures where only the MTG3 N-terminal domain (NTD) is bound, and the C-terminal domain has dissociated, are not at very high resolution. In spite of MTG3-FLAG purification being used, the putative NTD density could belong to some other protein. The sidechain densities for this region being unambiguous would have countered this possibility, but it is not clear if they are.

We agree with the reviewer that sidechain resolution would be desirable to unambiguously assign the density to the MTG3-NTD. Unfortunately, the resolution we attained in this region is not high enough to observe sidechain density for the MTG3-NTD. However, we believe this density can be assigned to the MTG3-NTD with reasonable confidence for several reasons. First, the density is entirely consistent with the location of the MTG3-NTD observed in previous, better resolved structures (Harper et al., 2023). Second, the trajectory of h44 is clearly altered compared to its mature state and is facing away from the foot region. This makes it highly unlikely that the density belongs to the lasso region of h44. To emphasize this, we now depict low-pass filtered maps in Figures 2 and 3, which show this more clearly (B1-G1, Fig 2 (g,i), Fig 3(d-g)). In combination with our biochemical data, we thus argue that it is justified to interpret this density as belonging to the MTG3-NTD. We have rephrased the text to more clearly emphasize the reasoning for this interpretation.

R3-5

5. The figures depicting structures seem excessively complicated. A jumble of colors makes it harder for readers to follow exactly what is being indicated. It is better to simplify them. For example, showing the ribosomal RNA can be dispensed with in multiple figure panels. Showing the full ribosome with multiple magnified views extracted within the same panel could be avoided. There might be other ways to avoid the visual clutter.

We have simplified the figures as suggested by the reviewer. In particular, we have removed rRNA depictions and changed the zoom-in views in several Figures. We hope they are now more clear and easier to interpret.

R3-6

6. Line 383, replace 'letter' with 'latter'.
Done.

Reviewer #3 (Remarks to the Author):

Fig. 2c: Location of boxes in the full small subunit images does not always seem to be the same as the enlarged panels d-i. The box labeled d has an extra vertical line.

We have corrected this.

Fig. 3c: Is it corresponding to the box in the full small subunit image in Fig. 2b?

h18 is shown in both in Fig 2b and Fig3c, however in a different context and from a different angle. We have narrowed the box in Fig2 and Fig3 to more clearly show what is highlighted (in Fig. 2b also h44 for instance).

The authors have made the changes required to address my previous comments. They have also modeled the states not previously modeled and deposited those models as well. I do have a few minor comments after examining the maps and models that they have kindly provided:

map A: 12SRNA residues helical hairpin residues 891-908 are not modeled but there seems to be density present for them, part of which shows up as unmodeled density even at high thresholds.

map B: There is unmodeled density near 12S rRNA residues helical hairpin residues 890-910. 12S rRNA h44 residues 1502-1548 not modeled, there is some additional clear double helical density that can be modeled.

map C: 12S RNA h44 residues 1502-1550 not modeled but there is some additional clear double helical density that can be modeled.

map D-F: Some extra poorly-resolved density on h44 visible that likely cannot be modeled accurately.

map G: Some extra poorly-resolved density on h44 and the mtIF2 domain interacting with tRNA. It may be worth providing a C-alpha model for that mtIF2 domain to show its overall position.

We thank the reviewer for his/her feedback on the models and aimed to address all the points mentioned. However, for some of the mentioned regions, we were not able to model individual residues with high enough confidence.

State A:

Although most of the additional density highlighted by the reviewer most likely belongs to immature h18, we can neither at high nor low threshold assign individual bases to the mostly distorted density due to its intrinsic flexibility (Figure 1A and C). Thus, we cannot model the region with high enough confidence and prefer to refrain from modelling those regions. Instead, we discuss the density as being part of h18 in the manuscript in Figure 1d, 2c, and Suppl. Fig. 5a-c.

However, we do agree that one could rigid-body fit the NTD of TFB1M from PDB 8CSP into a part of the density (Figure 1B), which we now did and updated the model accordingly.

We updated Fig 2c,d, Fig. 3b,c, Fig. 6, Tab. 1, and Suppl. Fig. 5c in the manuscript with some minor adjustments in the text (highlighted in cyan).

Figure 1

State B:

We agree with the reviewer that there is some additional unmodeled density belonging to the 12S rRNA present close to METTL15. This could belong to two residue regions of the 12S rRNA (1477-1489 or 1564-1572) both of which are in close proximity (Figure 2). However, the density does not show clear connectivity, making it impossible to clearly assign the unmodeled density. We therefore chose to refrain from modeling this region. Instead, we decided to discuss the density in the manuscript in Fig. 3f and Suppl. Fig. 4b.

Figure 2

State G:

We agree with the reviewer that it might be beneficial to show the overall position of the *mtlF2* domain that interacts with the tRNA. Since our map does have density for that region (Figure 3), we have now rigid-body fit the region taken from PDB 7PO2 and refined the B-factors against the low-resolution filtered map using phenix refine and updated the deposited model. We updated all figures showing the model for state G (Fig. 4b,g, Fig. 6, Tab. 1).

Figure 3

h44 in states B to G:

As requested by the reviewer we modelled additional residues (1498-1509, 1541-1551) of h44 into the density extending from the mtSSU in states B and C (Figure 4). We updated the deposited models and the manuscript accordingly (Fig. 2c,g,i, Fig. 3e, Fig. 6, Tab. 1, Suppl. Fig. 4a).

We would like to point out that the additional helical density is only clearly visible in unsharpened maps, which we decided to use as primary maps for state B and C to be able to better model the factors, which were less visible in the sharpened maps. In the (pre-) initiation states, the density for the factors was clearly visible in the sharpened maps, such that we chose sharpened maps as primary maps. Therefore, the density for unmaturing h44 is less prominent in state D to G and we cannot model additional residues of h44 with high enough confidence (Figure 5). We have also edited the methods section to clarify our use of sharpened and unsharpened maps.

Since we adjusted the models A, B, C, and G as part of the revision, we also updated the Suppl. Tab. 1 and 2 in the manuscript, the method section in the manuscript, and updated the submitted models in the PDB.

Figure 4

Figure 5

A with model

A

B

C

D

E

F

G